# Seeing Through the PRISM: Compound & Controllable Restoration of Scientific Images

**Rupa Kurinchi-Vendhan,**\* **Pratyusha Sharma, Antonio Torralba, Sara Beery**
Massachusetts Institute of Technology

## Abstract

Scientific and environmental imagery often suffer from complex mixtures of noise related to the sensor and the environment. Existing restoration methods typically remove one degradation at a time, leading to cascading artifacts, overcorrection, or loss of meaningful signal. In scientific applications, restoration must be able to simultaneously handle compound degradations while allowing experts to selectively remove subsets of distortions without erasing important features. To address these challenges, we present **PRISM** (**P**recision **R**estoration with **I**nterpretable **S**eparation of **M**ixtures). PRISM is a prompted conditional diffusion framework which combines compound-aware supervision over mixed degradations with a weighted contrastive disentanglement objective that aligns primitives and their mixtures in the latent space. This compositional geometry enables high-fidelity joint removal of overlapping distortions while also allowing flexible, targeted fixes through natural language prompts. Across microscopy, wildlife monitoring, remote sensing, and urban weather datasets, PRISM outperforms state-of-the-art baselines on complex compound degradations, including zero-shot mixtures not seen during training. Importantly, we show that selective restoration significantly improves downstream scientific accuracy in several domains over standard "black-box" restoration. These results establish PRISM as a generalizable and controllable framework for high-fidelity restoration in domains where scientific utility is a priority. Our code and dataset are available at this *link*.

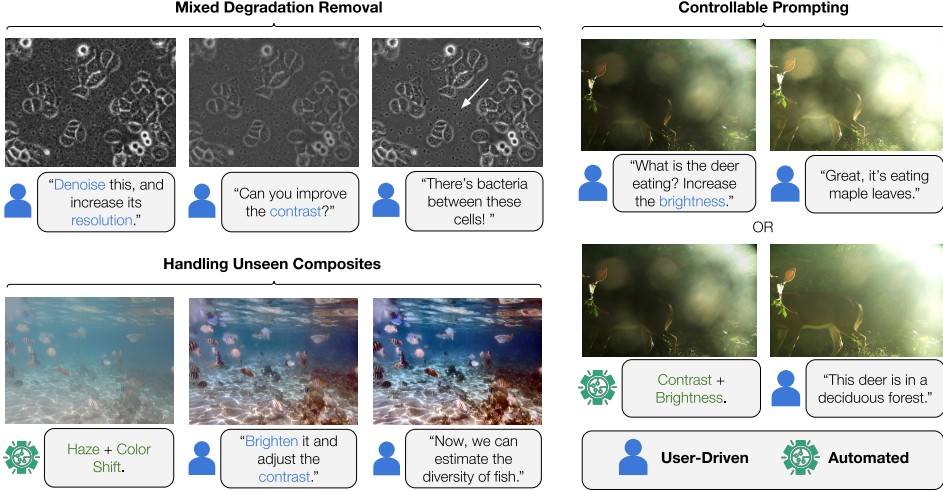

Figure 1: *Expert-in-the-Loop Restoration with PRISM.* PRISM enables robust compound restoration and zero-shot handling of unseen mixtures. It supports both automatic restoration and prompt-driven, selective correction for scientific analysis.

---

\*Corresponding author: rupak272@mit.edu

# 1 INTRODUCTION

Scientific and environmental imagery is rarely affected by a single degradation. Instead, images typically suffer from *compounding effects* that vary across datasets and collection settings: underwater images combine low light, scattering, and wavelength-dependent absorption effects (Akkaynak & Treibitz, 2018; Chiang & Chen, 2011), while satellite imagery suffers from overlapping sensor noise, haze, and cloud occlusions (King et al., 2013; Ahmad et al., 2019).

Specialized single-distortion models (e.g., for dehazing or removing clouds) enable domain experts to preprocess noisy data before conducting analysis. These approaches are often carefully hand-crafted for specific datasets and distortion types, making them brittle when degradations occur unpredictably, especially in scientific settings where ground truth is unavailable and the underlying distortions are not known a priori. For example, camera trap data may be affected by combinations of motion blur, weather, and lighting that can vary across images from the same deployment, making single-distortion pipelines ineffective. While this has motivated generalist "all-in-one" models (Li et al., 2020; Potlapalli et al., 2023b), current frameworks perform sequential/iterative removal of single distortions, which may lead to cascading artifacts and indiscriminately remove errors, erasing signals that should be preserved.

In scientific applications, restoration must preserve signals critical for precision and analysis, not just aesthetics. Aggressive restoration can inadvertently introduce more error: denoising may erase faint galaxies in astronomy data (Starck et al., 2002), and super-resolution can hallucinate subcellular structures in microscopy images (Christensen, 2022). Few models let experts control these tradeoffs.

We argue that restoration in scientific contexts requires three principles: *simultaneous over sequential correction*, *precision over aesthetics*, and in some cases, *control over automation*. We introduce PRISM, a conditional diffusion framework that simultaneously disentangles compound degradations and enables expert-guided, precision restoration. Figure 1 shows example use-cases of PRISM across scientific domains, demonstrating how our approach enables robust and interactive mixed degradation removal, how automated restoration can infer distortion types from an input image, and how expert prompts can selectively target distortions while preserving critical features. Our contributions are:

1. A **principled embedding design for compound degradations,** combining weighted contrastive learning with compound-aware supervision. The resulting **structured, compositional latent space** yields separable embeddings for primitives and their mixtures, enabling both automated compound restoration and selective distortion-specific control;

2. A **novel benchmark for scientific utility** spanning remote sensing, ecology, biomedical, and urban domains—including our newly-introduced Rooftop Cityscapes dataset—that reflects the needs of scientific workflows by evaluating task fidelity rather than perceptual quality;

3. A systematic study showing that **controllability is not a convenience but a necessity**. Selective, distortion-specific guidance significantly improves downstream scientific accuracy under compound degradations.

# 2 RELATED WORKS

## 2.1 RESTORATION IN SCIENTIFIC DOMAINS

Restoration has long been important for scientific imaging: early astronomy corrected photographic plates (Gull & Daniell, 1978), while biomedical imaging relied on denoising and deblurring for diagnostics (Buades et al., 2005; Dabov et al., 2007). Modern deep learning pipelines destripe astronomical surveys (Liu et al., 2025; Vojtekova et al., 2021) and denoise MRI images (Yan et al., 2024; Manjón & Coupe, 2018; Kidoh et al., 2020). While effective within their target domains, these methods typically assume that degradations are fixed, known, and well represented at training time.

In practice, scientific images often exhibit *compound* degradations. Domain-specific models such as Sea-Thru (Akkaynak & Treibitz, 2019) for underwater image correction explicitly model these effects, but rely on tailored, paired datasets which can limit their generalization when conditions deviate beyond their assumptions.

Additionally, cleaner data is not always better: scientific restoration often requires targeted improvements rather than blanket restoration. In microscopy, Lu et al. (2025) show that over-denoising can erase weak but biologically meaningful structures; in underwater monitoring, Cecilia & Murugan (2022) find that generic denoisers oversmooth fine-scale marine features critical for ecological interpretation. These findings motivate restoration frameworks that can handle compound effects while allowing experts to control *which* degradations are addressed.

## 2.2 HANDLING COMPOUND DEGRADATIONS

To address compound corruptions, several works propose shared-backbone or "all-in-one" architectures that jointly train on multiple degradation types, including All-WeatherNet (Li et al., 2020), TransRestorer (Chen et al., 2021), and SmartAssign (Wang et al., 2023). More recent universal models such as MT-Restore (Chen et al., 2022b), All-in-OneNet (Li et al., 2022b), and PatchDiffuser (Özdenizci & Legenstein, 2023) improve flexibility by expanding training coverage across degradation categories. These approaches use single-distortion removal for compositional restoration which may introduce unwanted artifacts or propagating errors from one stage to the next. Composite approaches, including OneRestore (Guo et al., 2024) and AllRestorer (Mao et al., 2024), explicitly model interactions between degradations rather than treating them independently, leading to improved performance on benchmark mixtures seen during training. Although these methods improve robustness to mixtures observed during training, they do not explicitly enforce a compositional latent structure that guarantees predictable behavior when selectively conditioning on subsets of degradations.

Research in disentangled representation learning suggests that factorized latent structures significantly improve generalization to novel combinations of known components (Burgess et al., 2018; Lake et al., 2017), particularly when architectures are designed to maintain the independence of these latent features as they transition through successive layers (Liang et al., 2025; Träuble et al., 2021). Our approach builds on this theory by aligning degradation mixtures with their primitives in the latent space to enable generalization to unseen data.

## 2.3 CONDITIONAL DIFFUSION AND PROMPT-GUIDED RESTORATION

Closely related methods on blind image restoration (BIR) handle cases where degradation types and severities are unknown at test time. Early GAN-based approaches such as BSRGAN (Zhang et al., 2021) and Real-ESRGAN (Wang et al., 2021) simulate diverse degradation pipelines to improve robustness under real-world conditions. Diffusion-based restoration more broadly extends conditional modeling with higher fidelity and stochastic control (Ho et al., 2020; Dhariwal & Nichol, 2021; Rombach et al., 2022). More recent diffusion-based BIR methods, including StableSR (Nagar et al., 2023) and DiffBIR (Lin et al., 2024), leverage pretrained diffusion priors to generate realistic and high-fidelity restorations without requiring explicit degradation labels. These models demonstrate strong perceptual quality and robustness to complex degradations. However, they rely on learned image priors and do not explicitly use semantic information from vision-language representations.

Prompt-guided approaches offer a more flexible interface for controlling restoration behavior. Methods such as PromptIR (Potlapalli et al., 2023b), demonstrate that conditioning on auxiliary text improves all-in-one restoration across a fixed vocabulary of degradation categories. DiffPlugin (Liu et al., 2024), MPerceiver (Ai et al., 2024), and AutoDIR (Jiang et al., 2024) introduce text- and image-conditioned diffusion models for universal image restoration. However, in these approaches, degradations are not explicitly disentangled in the model's latent space. As a result, removing one distortion may modify the image in unexpected ways, limiting reliable partial restoration. We distinguish between prompt-conditioned restoration and structurally controllable restoration: the former allows specifying tasks, whereas the latter requires a representation in which degradation factors are disentangled enough to support predictable, selective intervention.

In parallel, several works adapted vision–language models for restoration by fine-tuning CLIP representations to be degradation-aware. DA-CLIP aligns image embeddings with textual degradation descriptions, improving robustness and downstream restoration performance under domain shift (Luo et al., 2023; 2024). AutoDIR further introduces a semantic-agnostic loss that encourages CLIP to distinguish clean from degraded images based on quality-related cues rather than semantic content, biasing the encoder toward degradation-sensitive features (Jiang et al., 2024). These works primarily

focus on aligning representations to individual distortion types rather than enforcing compositional structure that supports systematic generalization to unseen mixtures.

A recent survey by Jiang et al. (2025) highlights limited support for real-world compound degradations and appropriate evaluation protocols. Existing methods also lack compositional structure for stable joint correction and distortion-specific control. We address these gaps by modeling degradations compositionally and introducing a mixed degradations benchmark and a downstream scientific task evaluation, which together assess structural generalization and practical utility under realistic compound settings.

## 3 METHODS

PRISM combines compound-aware supervision with contrastive latent disentanglement to enable robust and controllable restoration under mixed degradations.

### 3.1 BUILDING A DATASET OF COMPOUND DEGRADATIONS

We construct a synthetic dataset from diverse scientific domains: ImageNet (Deng et al., 2009), Sentinel-2 patches from Sen12MS (Schmitt et al., 2019), iWildCam 2022 (Beuving et al., 2022), EUVP underwater imagery (Islam et al., 2020), CityScapes (Cordts et al., 2016), BioSR microscopy slides (Gong et al., 2021), Brain Tumor MRI (Nickparvar, 2021), and high-resolution Subaru/HSC sky surveys (Miao et al., 2024). Across these datasets, we sample 2M clean images that serve as the ground truth targets during training. See Appendix B for more details.

**Compound-Aware Supervision.** Each image is degraded by up to three distortions sampled from a library including geometric warping, blur, photometric shifts and weather effects, etc. We apply a maximum of three distortions per image to capture challenging compound cases while maintaining enough of the original semantic content (see Table 9 in Appendix E). Each image is distorted by a composition of multiple augmentations.

Following prior work such as Real-ESRGAN (Wang et al., 2021), which demonstrate that increased variability in training degradations can improve robustness and generalization, we construct a diverse spectrum of degradations to better approximate real-world conditions (Luo et al., 2024; 2022; Zhang et al., 2021). The distortions are applied in random order with varied parameters, such as kernel size for blurring or angle of snowfall, that determine degradation intensity. We use GPT-4 (Hurst et al., 2024) to generate variable natural language prompts describing image degradations to simulate realistic user instructions (e.g., "remove haze," "dehaze this image," "dehaze and super-resolve this"). We also include *partial* prompts (remove a subset of distortions) and *negative* prompts (remove a non-present distortion). This design is critical for selective restoration: by exposing the model to submixtures and negative conditions, we encourage it to associate each degradation primitive with a distinct latent direction and to avoid unintended corrections when a distortion is not specified. This supervision explicitly supports predictable, distortion-specific intervention at inference time. Overall, the dataset consists of triplets $(I_{\text{clean}}, I_{\text{dist}}, p)$. Further details on dataset sampling and the distortion pipeline are provided in Appendix B.

### 3.2 THE PRISM MODEL

Our framework builds on composite/all-in-one restoration (Guo et al., 2024; Jiang et al., 2024; Ai et al., 2024) but emphasizes *controllability* and *precision* under compound degradations. Figure 2 outlines our process: we first fine-tune the CLIP image encoder on our mixed degradation dataset to ensure that embeddings preserve semantic content while becoming distortion-invariant, keeping the text encoder frozen. Once adapted, we freeze both CLIP encoders to provide a stable conditioning space for training the latent diffusion backbone.

**Stage 1: Disentangling Distortions.** Naive CLIP embeddings are poorly suited for restoration, as they primarily cluster images by semantic content rather than degradation type or quality (Radford et al., 2021). Prior work (Jiang et al., 2024) showed that quality-aware embeddings improve performance by shifting focus from semantics to distortions. We extend this to compound degradations by explicitly modeling *compositionality*, ensuring embeddings reflect the presence and overlap of

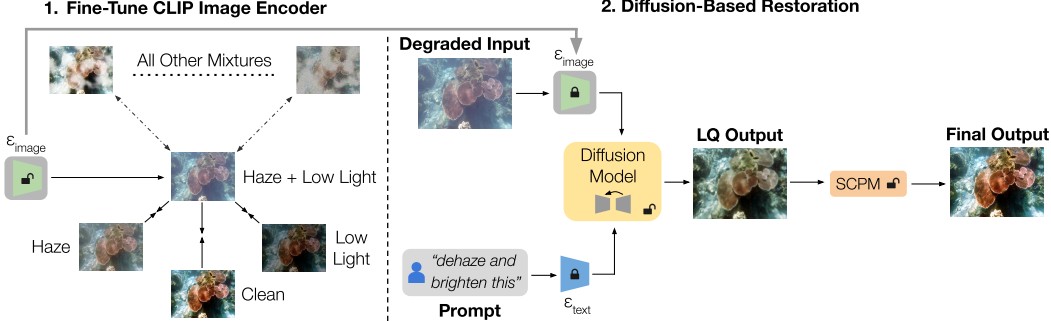

Figure 2: *Overview of PRISM.* We first fine-tune CLIP's image encoder to disentangle image embeddings by distortion. The degraded input and user prompt are then used to condition the latent diffusion backbone, the coarse outputs of which are refined with a Semantic Content Preservation Module (SCPM) to yield the final restored output. Appendix Figure 12 shows the SCPM's architecture.

multiple distortions (e.g., an image degraded by haze+rain should be more similar to haze-only and rain-only images than to images degraded by unrelated distortions such as noise).

Let $I_{\text{clean}}$ be a clean image and $\{I_{\text{dist}}^{(j)}\}_{j=1}^{m}$ its $m$ degraded variants. For cosine similarity $\text{sim}(\cdot, \cdot)$, we use a contrastive loss aligning the corresponding embeddings $e_{\text{dist}}^{(j)}$ with $e_{\text{clean}}$ and repelling it from its $m-1$ sibling variants and all variants from other images in the batch. To encode the similarity structure of different combinations of distortions, we use the Jaccard distance (Jaccard, 1901) between their distortion sets $d^{(j)}$ and $d^{(k)}$

$$w_{jk} = \exp\left(1 - \frac{|d^{(j)} \cap d^{(k)}|}{|d^{(j)} \cup d^{(k)}|}\right).$$

We analyze this weighting mechanism in Appendix E. The resulting per-variant loss is

$$\mathcal{L}_{\text{ctr}}^{(j)} = -\log \frac{\exp(\text{sim}(e_{\text{dist}}^{(j)}, e_{\text{clean}})/\tau)}{\sum_{k \neq j} w_{jk} \cdot \exp(\text{sim}(e_{\text{dist}}^{(j)}, e_{\text{dist}}^{(k)})/\tau) + \sum_{l \in \mathcal{B}_{\text{other}}} \exp(\text{sim}(e_{\text{dist}}^{(j)}, e_{\text{other}}^{(l)})/\tau)}.$$

for $\tau = 0.10$. We use $\mathcal{B} = 256$ clean images per batch, each with $\mathcal{B}_{\text{other}} = 256$ randomly sampled degraded variants from our library of primitive and compound distortions.

The contrastive objective encourages degraded embeddings to reflect distortion structure, but it does not explicitly prevent the clean embedding from drifting toward degradation-sensitive features. We introduce a quality-aware regularizer that penalizes the clean embedding for exhibiting distortion evidence

$$\mathcal{L}_{\text{qual}}^{(j)} = \sum_{c \in d^{(j)}} \hat{p}(c \mid e_{\text{clean}}),$$

where $\hat{p}(c \mid e_{\text{clean}})$ is the predicted probability of distortion $c$ from $e_{\text{clean}}$. This term discourages degradation hallucination and anchors the representation to remain distortion-free for clean inputs.

The final encoder loss is $\mathcal{L}_{\text{CLIP}} = \frac{1}{m} \sum_{j=1}^{m} (\mathcal{L}_{\text{ctr}}^{(j)} + \mathcal{L}_{\text{qual}}^{(j)})$, yielding embeddings that: (i) align degraded and clean images, (ii) reflect compositional overlap, and (iii) support controllable restoration.

**Stage 2: Restoring with Conditional Diffusion.** We adopt a latent diffusion model, Stable Diffusion v1.5 (Rombach et al., 2022). During training, we replace the standard CLIP image encoder with our fine-tuned compound distortion-aware encoder, providing semantic and degradation context. Then, we concatenate the conditioning vector from the image encoder with the text embedding of the restoration prompt from the frozen CLIP text encoder. These embeddings are passed through a learnable cross-attention layer at each UNet block following the original Stable Diffusion v1.5 design for prompt conditioning. This allows PRISM to attend to the desired set of degradations during each denoising step. Unlike sequential approaches (Jiang et al., 2024), PRISM conditions on the full composite prompt in a single denoising trajectory, enabling joint restoration of overlapping artifacts. While MPerceiver (Ai et al., 2024) encodes multiple degradations as concatenated tokens, this strategy

represents multi-distortion inputs but does not explicitly enforce disentangled or compositional structure in the embedding space that supports controlled recombination.

Latent diffusion can blur fine-scale structures critical in scientific imagery. Following Jiang et al. (2024), we integrate a Semantic Content Preservation Module (SCPM), a lightweight decoder-side refinement block that adaptively fuses encoder and decoder features to preserve edges and small textures. Full architectural details and ablations are provided in Appendix E.

For fair comparison, all baselines are trained on the fixed set of primitive distortions. Training details and compute requirements are provided in Appendix A, baselines are described in Appendix D, and ablations over model components are provided in Appendix E.

### 3.3 PROMPTING

PRISM enables both *expert-guided* and *automated* restoration by grounding free-form natural language prompts in a structured task space.

At inference time, PRISM supports:

- **Expert-guided restoration:** A free-form instruction (e.g., "remove fog and color shift") is embedded with the frozen CLIP text encoder and used to condition diffusion.

- **Automated restoration:** Given an input image, a lightweight MLP predicts a multi-label distortion set from the image embedding. This set is converted into a standardized prompt ("remove distortions $x, y, z$") and encoded by CLIP.

In both prompting scenarios, the text embeddings condition the latent diffusion model through cross-attention at each denoising step. Sensitivity to prompt variation is analyzed in Appendix E.

### 3.4 EVALUATION

We evaluate PRISM on: (1) compound and controllable restoration, (2) handling unseen real-world composites, and (3) downstream utility. Unless noted otherwise, we evaluate manual restoration with predefined distortion types, using the free-form prompts generated by GPT-4. Full details on datasets and evaluation are in Appendix B and C.

**Mixed Degradations Benchmark (MDB).**   We use the held-out subset of the triplets $(I_{\text{clean}}, I_{\text{dist}}, p)$ from our dataset as our fixed testbed. This MDB measures sequential vs. composite prompting and prompt faithfulness under compound degradations. This dataset extends beyond the CDD-11 dataset proposed by Guo et al. (2024) to span a broad set of real-world degradations, with varied intensities.

**Handling Unseen Distortions.**   For zero-shot tests, we evaluate on real domains with compound distortions not explicitly seen in training: underwater effects in UIEB (Li et al., 2019), under-display camera artifacts (Zhou et al., 2021), and fluid distortions (Thapa et al., 2020). These datasets probe PRISM's ability to extend to novel distortions.

**Downstream Utility.**   Standard benchmarks measure pixel similarity to a clean reference, but such metrics fail to capture whether restored images remain scientifically useful. We instead evaluate restoration through downstream tasks using real datasets with natural distortions and undistorted views as ground truth. To reflect how restoration outputs are typically used in practice, we use off-the-shelf pretrained models, giving a conservative but practical measure of utility. We test across four domains:

1. **Remote sensing with Sen12MS (Schmitt et al., 2019):** landcover classification (Papoutsis et al., 2023) on cloudy satellite data, with labels from cloudless samples.

2. **Wildlife monitoring with iWildCam 2022 (Beuving et al., 2022):** species classification with SpeciesNet (Gadot et al., 2024) on low-confidence nighttime images with expert labels from high-confidence frames of the same sequence.

3. **Segmentation and fluorescence in microscopy with BioSR (Gong et al., 2021):** segmentation and fluorescence/intensity measurements of clathrin-coated pits from low signal-to-noise

data using MicroSAM (Archit et al., 2025), compared to high quality structured illumination microscopy (SIM) ground truth.

4. **Urban forest monitoring using our novel Rooftop Cityscapes dataset:** panoptic segmentation (Lin et al., 2017) of cityscapes under haze/low light, with paired, labeled clear-weather data. See Appendix C for details on this custom dataset.

# 4 RESULTS AND DISCUSSION

## 4.1 BREAKING THE CASCADE: COMPOUND RESTORATION MADE ROBUST

Table 1: PRISM outperforms baselines with manual prompting on MDB, where each test image has up to three distortions. Best results are **bolded**, second-best are underlined.

| Category | Method | PSNR ↑ | SSIM ↑ | FID ↓ | LPIPS ↓ |
|---|---|---|---|---|---|
| All-in-One | AirNet | 14.76 | 0.742 | 78.55 | 0.382 |
| | Restormer$_A$ | 16.32 | 0.768 | 70.11 | 0.365 |
| | NAFNet$_A$ | 16.98 | 0.776 | 68.30 | 0.352 |
| | PromptIR | 18.11 | 0.801 | 62.78 | 0.298 |
| Diffusion | DiffPlugin | 19.07 | 0.821 | 53.88 | 0.255 |
| | MPerceiver | 20.84 | 0.829 | **48.18** | 0.235 |
| | AutoDIR | 20.42 | 0.833 | 50.75 | 0.246 |
| Composite | OneRestore | 19.36 | 0.812 | 59.42 | 0.276 |
| | **PRISM (ours)** | **22.08** | **0.842** | 48.97 | **0.218** |

Sequentially removing distortions often accumulates errors, artifacts, or inconsistencies, while restoring all distortions jointly yields more stable, high-fidelity results. Our MDB evaluation supports this intuition, with qualitative results in Appendix Figs. 17 and 16.

Table 1 highlights a divide between early all-in-one models (Li et al., 2022a; Zamir et al., 2022; Chen et al., 2022a; Potlapalli et al., 2023a), which are trained per distortion and generalize poorly to mixtures, recent diffusion approaches Liu et al. (2024); Ai et al. (2024); Jiang et al. (2024), and models that explicitly target composite restoration (Guo et al., 2024). While OneRestore is trained on composite datasets like PRISM, it fails to match the perceptual quality of diffusion model outputs; however, these diffusion methods typically remain limited by their reliance on single-distortion or sequential training regimes.

PRISM achieves the best results across both fidelity (PSNR/SSIM) and perceptual metrics (FID/LPIPS), owing to two design choices: (1) *compound-aware supervision* over mixed degradations, and (2) *contrastive disentanglement* of distortions. We study the contributions of each below.

**Compound-aware supervision supports restoration under increasingly complex mixtures.** Training on combinatorial mixtures of degradations (full, partial, and negative restoration) allows simultaneous removal of multiple effects without cascading errors. Fig. 3 shows that while PRISM matches baselines on images with a single degradation, it significantly outperforms them as complexity grows, especially on unseen cases with four distortions (an extension of the MDB set). Training on composites explicitly outperforms training on primitives separately, and that improved image embeddings from our contrastive loss provide an additional boost over baselines. Importantly, this compound-aware structure not only improves robustness under increasing distortion complexity, but also lays the foundation for selective restoration. These trends hold consistently across other metrics (SSIM, LPIPS, and FID), as shown in Appendix Fig. 14).

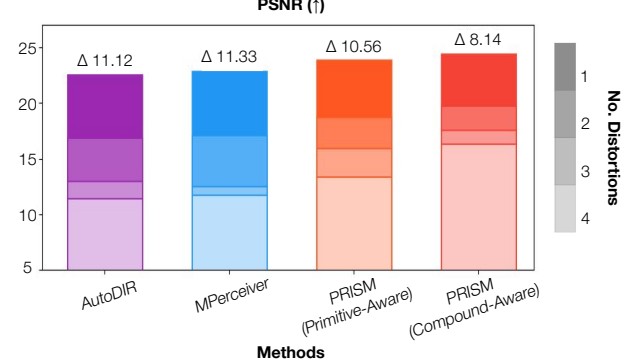

Figure 3: *PRISM trained on composite examples scales best with the number of distortions.* This outperforms our model trained on each degradation separately as well as comparable baselines, emphasized by the Δ PSNR of test images with 1 vs. 4 distortions) above each bar.

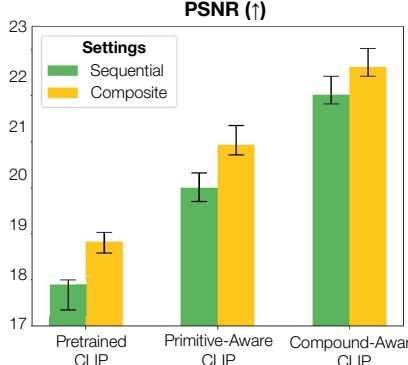

Figure 4: *Latent disentanglement enables both stepwise and single-shot restoration, closing the gap between prompting strategies.*

**Contrastive disentanglement improves compositional restoration.** Appendix Fig. 13 visualizes the effect of the fine-tuning CLIP on the image embedding space. Our compound-aware weighted contrastive loss explicitly incorporates mixed degradations, enforcing a latent geometry that pulls compound distortions toward the span of their primitives (right). This design closes the gap between sequential and single-shot prompting (Fig. 4), ensuring that multi-step restoration behaves predictably and that single-shot prompts achieve comparable fidelity. Appendix Fig. 15 demonstrates that these improvements generalize beyond PSNR. This allows experts to target specific degradations without disturbing other content. By combining compound-aware supervision with contrastive disentanglement, PRISM not only outperforms existing baselines on restoring compound degradations, but also maintains fine control.

## 4.2 COMPOSITIONALITY ENABLES ADAPTIVE RESTORATION IN NOVEL SETTINGS

The compositional structure of the image latent space also supports generalization to unseen degradations. If degradations are represented compositionally, then novel composites can be modeled as combinations of known primitives. This means PRISM can automatically identify constituent distortions and restore them, even if the exact combination was never seen in training.

We evaluate zero-shot restoration on three domains (underwater imagery, under-display cameras, and fluid lensing), each containing complex, previously unseen distortions (Table 2). For each dataset, we use the compound-aware CLIP encoder to identify the fixed set of distortion types present in the images of each dataset. We then apply the same manual prompts over this standardized set for all models to ensure a fair, consistent evaluation. While the predicted distortion categories for UIEB were more variable and often reflected mixtures of multiple effects such as low light, haze, contrast, and color shifts, the POLED and ThapaSet datasets exhibited more uniform and stable classifications, providing clearer mappings to their dominant distortion types: low light, blur, and contrast vs. refraction and warping, respectively. Qualitative examples are shown in Appendix F.

Table 2: PRISM achieves state-of-the-art zero-shot performance across underwater (UIEB), under-display camera (POLED), and fluid lensing (ThapaSet) benchmarks. Best results are **bolded**, second-best are underlined.

| Category | Method | UIEB | | | POLED | | | ThapaSet | | |
|---|---|---|---|---|---|---|---|---|---|---|
| | | PSNR ↑ | SSIM ↑ | LPIPS ↓ | PSNR ↑ | SSIM ↑ | LPIPS ↓ | PSNR ↑ | SSIM ↑ | LPIPS ↓ |
| All-in-One | AirNet | 17.51 | 0.768 | 0.468 | 13.61 | 0.582 | 0.529 | 18.41 | 0.388 | 0.609 |
| | Restormer$_A$ | 18.13 | 0.792 | 0.454 | 14.38 | 0.608 | 0.512 | 19.31 | 0.413 | 0.588 |
| | NAFNet$_A$ | 17.76 | 0.756 | 0.479 | 13.02 | 0.591 | 0.543 | 18.94 | 0.401 | 0.596 |
| | PromptIR | 19.76 | 0.858 | 0.417 | 16.42 | 0.651 | 0.468 | 20.63 | 0.439 | 0.564 |
| Diffusion | DiffPlugin | 20.72 | 0.874 | 0.392 | 17.01 | 0.659 | 0.451 | 21.15 | 0.454 | 0.536 |
| | MPerceiver | 21.18 | 0.889 | 0.366 | 17.55 | 0.669 | 0.436 | 21.41 | 0.459 | 0.522 |
| | AutoDIR | 21.02 | 0.887 | 0.374 | 17.33 | 0.665 | **0.431** | 21.53 | 0.462 | 0.528 |
| Composite | OneRestore | 20.53 | 0.869 | 0.404 | 16.82 | 0.654 | 0.457 | 20.82 | 0.446 | 0.548 |
| | **PRISM (ours)** | **22.18** | **0.914** | **0.331** | **18.26** | **0.694** | 0.419 | **22.36** | **0.487** | **0.492** |

Both all-in-one models like AirNet and Restormer and composite methods such as OneRestore fall short in capturing complex degradation interactions without robust priors. In contrast, diffusion models like AutoDIR generalize well but often treat distortions in isolation, which may introduce unwanted intermediate errors in linearly restoring inputs.

PRISM's state-of-the-art zero-shot performance is driven by a latent space that treats compound degradations as structured combinations of primitives. Rather than attempting to model the complex, non-linear physics of every possible distortion interaction, the contrastive design enforces a compositional logic. By training on submixtures, the model learns to represent a composite (e.g., haze + overexposure) as a joint embedding of its constituent parts. This allows the model to map previously unseen mixtures into a known coordinate system defined by established primitives, effectively "in-

terpolating" a restoration strategy for novel compounds. Consequently, PRISM avoids the pitfalls of brittle category memorization, enabling the restoration of unseen mixtures using the semantic proximity between known primitives and unknown composites without additional supervision.

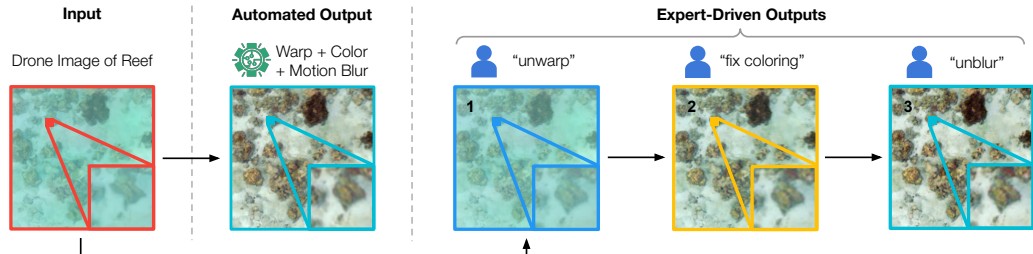

Figure 5: *Structured, compositional latent geometry supports both automated (left) and expert-driven (right) generalization.* With an expert-in-the-loop, prompts progressively target distortions.

Fig. 5 illustrates how this design translates into practice with high-resolution drone imagery over coral reefs in Mo'orea (Saccomanno et al., 2025). While our method enables automated classification and correction of novel distortions, it also provides interpretability and control: experts can explore how distortions relate, refine restoration strategies, and avoid black-box corrections. In improving performance on partial restoration, PRISM enables experts to issue stepwise prompts (e.g., "unwarp," "fix coloring," "unblur") to iteratively restore unseen distortions as they see fit.

### 4.2.1 THE CASE FOR CONTROLLABILITY IN SCIENCE

While PRISM restores images robustly under compound degradations, we argue that in some applications, restoration quality cannot be judged by appearance alone. We assess PRISM's impact on downstream tasks across four domains: remote sensing, ecology, microscopy, and urban monitoring.

This raises a critical question: if a model can remove all degradations, should it? In scientific imaging, often *no*. Distortions can be mixed with faint but meaningful signals, and indiscriminate restoration may erase these cues or introduce artifacts that mislead downstream models. Maintaining downstream fidelity requires control: experts must choose what to correct versus preserve.

As shown in Table 3, controllability significantly improves downstream performance over full restoration (automatically detecting all distortions present) in three of four domains. For instance, in nighttime camera trap data, restoring only contrast may improve recognition over full restoration, which can blur subtle texture cues. In urban scenes, removing haze improves segmentation, but also brightening the image may over-adjust vegetation in the distance. Remote sensing is the exception: full, automatic restoration performs slightly better, as removing only clouds leaves images under-illuminated and hazy.

Table 3: Selective controllability outperforms full restoration across three of four downstream tasks. We report mean $\pm$ std over 3 random seeds. Best results are **bolded**.

| Domain | Degraded Input | Full Restoration | Selective Restoration | p-value |
|---|---|---|---|---|
| Remote sensing (Acc. ↑) | $0.781 \pm 0.013$ | $\mathbf{0.842 \pm 0.011}$ | $0.836 \pm 0.012$ | 0.11 (n.s.) |
| Camera Traps (Acc. ↑) | $0.921 \pm 0.004$ | $0.976 \pm 0.008$ | $\mathbf{0.984 \pm 0.004}$ | $0.032 < 0.05$ |
| Microscopy (mIoU ↑) | $0.353 \pm 0.015$ | $0.475 \pm 0.012$ | $\mathbf{0.580 \pm 0.010}$ | $0.018 < 0.05$ |
| Urban scenes (mIoU ↑) | $0.548 \pm 0.018$ | $0.615 \pm 0.014$ | $\mathbf{0.650 \pm 0.012}$ | $0.041 < 0.05$ |

Fig. 6 illustrates a use case of controllability in microscopy: super-resolution alone improves segmentation alignment with SIM ground truth, but additional denoising erases faint, biologically relevant structures. Additional examples across other domains are included in Appendix F.

In some cases, *controllability enables different analyses on the same data* which may depend on fundamentally different visual cues. Within microscopy, segmentation of clathrin-coated pits relies on high-frequency structural detail, such as sharp boundaries and local contrast that delineate subcellular objects. In contrast, fluorescence measures the mean pixel intensity within biological regions, a proxy for molecular concentration or protein recruitment. This calculation is highly sensitive to intensity bias and noise variance but relatively tolerant to mild blur.

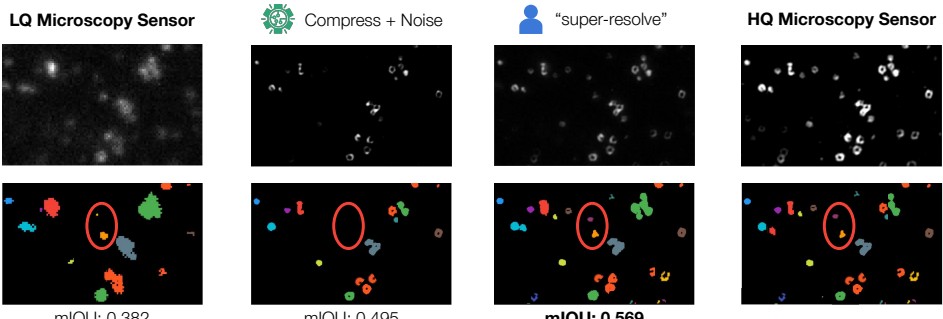

Figure 6: *Selective restoration improves segmentation of clathrin-coated pits in microscopy.* Super-resolution alone improves mIoU, while automatically detecting and removing noise suppresses faint but biologically relevant signals (see regions encircled in red), reducing accuracy.

Table 4 summarizes performance across restoration settings. Super-resolution yields the best segmentation mIoU by enhancing edges and subcellular boundaries, but this same sharpening increases fluorescence error. Denoising has the opposite effect. While it preserves the intensity distribution of input data, producing the lowest fluorescence MSE, it removes fine structure needed for accurate segmentation. Combined restoration produces intermediate or degraded results, demonstrating that no single transformation can satisfy both scientific objectives simultaneously.

These findings highlight that *restoration is task-dependent*, and experts may not always know how to preprocess their data beforehand. Blanket restoration can overlook the varying demands of scientific workflows. PRISM's controllability enables experts to ensure that restored images remain faithful to the underlying scientific signal as needed.

Important challenges remain for PRISM. Our training still depends on synthetic augmentations that cannot fully capture real distortions. Moreover, extending controllability beyond specifying which distortions to remove to their intensity and spatial extent would enable localized restoration and finer-grained preservation of scientific signals.

Table 4: *Task-dependent restoration performance on BioSR*. Segmentation prefers structural enhancement, whereas fluorescence quantification requires accurate intensity preservation. No single restoration setting is optimal for both tasks.

| Restoration Prompt | Segmentation mIoU ↑ | Fluorescence MSE ↓ |
|---|---|---|
| Denoise | 0.479 | **0.006** |
| Super-Resolution | **0.580** | 0.018 |
| Combined | 0.475 | 0.014 |
| No restoration | 0.356 | 0.025 |

While diffusion-based restoration typically demands greater computational resources than traditional encoder–decoder architectures, PRISM maintains competitive runtimes. As detailed in Appendix E (Table 13), its latency is comparable to existing state-of-the-art diffusion frameworks. In this way, PRISM balances practical efficiency with addressing key priorities in scientific restoration.

## 5 CONCLUSIONS

Our results show that controllable *and* compound-aware restoration is critical for scientific and environmental imaging. PRISM outperforms both specialized and generalist baselines by combining (1) compound-aware supervision, which exposes the model to overlapping degradations, and (2) weighted contrastive disentanglement, which organizes the latent space according to composite distortions. Together, these yield more robust and interpretable restoration.

We also find strong generalization beyond curated training sets. PRISM achieves robust zero-shot performance on underwater imaging, under-display camera correction, and fluid lensing, showing that compositional representations extend naturally to unseen domains. Importantly, evaluations on real composite degradations confirm generalization beyond our synthetic training pipeline.

A key insight is that *more restoration is not always better*. Across diverse domains, we show that indiscriminate removal of degradations suppresses faint but meaningful signals or introduces artifacts. Moreover, the appropriate level and type of restoration can be *task-dependent*, as different scientific analyses may rely on distinct visual cues and therefore benefit from different restoration strategies. Allowing experts to choose which distortions to correct is essential for scientific precision.

ACKNOWLEDGMENTS

This work was supported in part by a Schmidt Science's AI2050 Early Career Fellowship, NSF CAREER Grant (Award No. 2441060), an NSF Graduate Research Fellowship (Award No. DGE-2141064), the NSF and NSERC AI and Biodiversity Change Global Center (NSF Award No. 2330423 and NSERC Award No. 585136), and the MIT Generative AI Consortium.

ETHICS STATEMENT

This work relies exclusively on publicly available datasets and synthetic distortions; all source data are cited appropriately. We acknowledge that restoration models can be misapplied or introduce misleading artifacts, and note that our system is not intended for high-stakes decision-making without domain expert oversight, but as an overall framework design for handling complex real-world data.

REPRODUCIBILITY STATEMENT

Our code repository includes reproducible pipelines for data generation, model training, inference, and evaluation across standard and downstream benchmarks. Detailed descriptions of the dataset preprocessing, model architecture, hyperparameters, training objectives, and evaluation protocols are included in the main paper and appendix, along with extensive ablation studies and full experimental results. All baselines use publicly available implementations following the original authors' specifications, and we additionally release our benchmarking dataset for mixed degradation removal and evaluation protocols for all downstream task analysis.

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
