## A    TRAINING DETAILS

PRISM is trained on $8 \times 40$GB NVIDIA A100 GPUs using mixed-precision training (FP16).

**Hyperparameters.**    For our final model we use:

- Batch size: 264 (global, distributed across 8 GPUs)
- Learning rate: $1 \times 10^{-4}$ (AdamW optimizer)
- No learning-rate warm-up; cosine decay schedule
- Training epochs: 500
- Input resolution: $256 \times 256$
- Gradient clipping: 1.0
- EMA decay: 0.9999 for model weights

We initialize the backbone from publicly available Stable Diffusion v1.5 weights (Rombach et al., 2022).

**CLIP Fine-Tuning.**    For the embedding space, we initialize from OpenAI CLIP ViT-B/32 (Radford et al., 2021) pretrained weights. We fine-tune only the final projection layers and cross-attention adapters, freezing the base vision and text encoders to preserve semantic alignment. Fine-tuning uses:

- Batch size: 256
- Learning rate: $5 \times 10^{-5}$ (AdamW optimizer)
- Training epochs: 50

**Distortion Classifier for Automated Restoration.**    We train the distortion classifier after fine-tuning CLIP, using the frozen distortion-aware CLIP image encoder as a feature extractor. The two-layer MLP has a hidden dimension of 512 and predicts over a fixed vocabulary of $K = 14$ primitive distortions (blur, haze, low light, color shift, rain, snow, noise, etc.).

We train the classifier using the same synthetic mixed-degradation dataset used to fine-tune the CLIP image encoder, treating each degraded image $I$ as a multi-label instance with ground-truth distortion set $\mathcal{D} \subseteq \{1, \ldots, K\}$. The label set is converted into a binary target vector $\mathbf{y} \in \{0, 1\}^K$, and we optimize a binary cross-entropy loss. We use parameters

- Batch size: 512
- Learning rate: $1 \times 10^{-4}$ (AdamW optimizer)
- Training epochs: 98
- Cosine learning-rate decay schedule

The CLIP image encoder is kept frozen during this stage; only the MLP parameters are updated. At test time, we predict the distortion set as $\hat{d} = \{\, k : \hat{p}_k > 0.85 \,\}$ and convert $\hat{d}$ into the associated standardized prompt "remove the effects of distortions $d_1, d_2, \ldots, d_m$".

## B    MDB CONSTRUCTION

As stated in Section 3, our composite degradation dataset used for ground truth during training was drawn from a diverse collection of datasets spanning multiple scientific and environmental imaging domains. Table 5 summarizes the datasets used.

Each clean image is transformed into multiple distorted counterparts so that every distortion type is applied across the full dataset. For each image, we apply $N$ distortions. We limit $N$ to a maximum of 3 distortions because we found that $N \geq 4$ degraded the signal significantly and made the task of restoration too difficult (see Appendix E for the relevant ablation over $N$).

Table 5: Summary of Training Datasets. PRISM is trained on a diverse set of natural and scientific domains spanning ecological, medical, astronomical, and remote sensing imagery.

| Dataset | Description | Size |
|---|---|---|
| **ImageNet** (Deng et al., 2009) | 1000-class benchmark of natural images with visually diverse scenes. | 1.2M |
| **Sen12MS (Sentinel-2)** (Schmitt et al., 2019) | RGB satellite image patches with and without clouds, used for land-cover and cloud-removal tasks. | 720K |
| **iWildCam 2022** (Beuving et al., 2022) | Camera trap sequences for wildlife monitoring under challenging lighting and environmental conditions. | 28.8K |
| **EUVP** (Islam et al., 2020) | Paired underwater photos with clear vs. distorted conditions (enhanced clean split). | 3.7K |
| **Cityscapes** (Cordts et al., 2016) | Urban street scenes captured from vehicle-mounted cameras (includes 5K labeled and 20K "extra"). | 25K |
| **BioSR** (Gong et al., 2021) | Fluorescence microscopy slides (wide-field vs. SIM ground truth) for super-resolution and denoising tasks. | 14K |
| **Brain Tumor MRI** (Nickparvar, 2021) | Clinical MRI scans for tumor detection and segmentation with paired clean/noisy modalities. | 7K |
| **AstroSR HSC Surveys** (Miao et al., 2024) | Wide-field sky survey images from the Hyper Suprime-Cam (HSC), used for astrophysical source recovery. | 2K |

Given the sampled number $N$, distinct distortion types are drawn uniformly from a predefined library $\mathcal{D}$ of transformations. Our distortion set spans 14 categories: including geometric distortions (motion blur, warping, refraction, defocus blur), photometric degradations (contrast, color shifts, brightness, low light), occlusions (clouds, haze, rain, snow), and noise-based effects (additive noise, compression).

**Geometric Distortions.** *Elastic Deformation* generates random displacement fields from $\mathcal{U}(-1,1)$, applies Gaussian smoothing ($\sigma \in [20,30]$), scales by $\alpha \in [10,20]$, and warps images via bilinear interpolation. *Refraction* uses Gaussian-filtered noise ($\sigma = 10$) with strength $\in [20,80]$ applied to coordinate maps. *Motion Blur* creates directional kernels (size $\in [5,10]$) along horizontal, vertical, or diagonal axes, with optional depth-aware masking that selectively blurs foreground or background regions based on threshold $\tau \in [0.3,0.7]$.

**Photometric Distortions.** *Low Light* applies multiplicative brightness reduction with factor $f \in [0.4,0.9]$. *Color Jitter* combines independent RGB channel shifts ($\delta \in [-0.4,0.4]$) with

**1. Sample the number of distortions to apply**

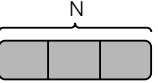

**2. Sample the N distortions**

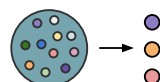

**3. Sample the parameters per distortion**

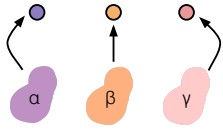

**4. Randomize the order**

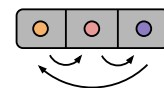

Figure 7: An overview of the data augmentation pipeline of diverse compound degradations.

color casts (warm/cool/green/magenta/cyan/yellow) at intensity $\alpha \in [0.1,0.3]$. *Overexposure* uses inverse gamma correction ($\gamma = 1/f$, $f \in [1.0,1.5]$), brightness boost, and soft highlight clipping above threshold $\tau \in [0.4,0.9]$. *Underexposure* applies gamma with $f \in [0.5,0.9]$, shadow compression below $\tau \in [0.1,0.3]$, and luminance-weighted noise ($\sigma \in [0.02,0.08]$) in dark regions. *Contrast* and *Saturation* use random factors from $[0.4,1.0]$.

**Occlusion Effects.** *Haze* blends images with white layers using depth-scaled opacity $\alpha \in [0.65,0.9]$: $I_{haze} = I(1-D\alpha)+D\alpha$. *Rain* generates two-scale streaks (blur kernels $\in \{7\text{-}23\}$, zoom $\in [1.0,3.5]$) with selective fog (90th percentile depth, visibility $[8000,15000]$) and opacity $\alpha \in [0.2,0.4]$. *Snow* uses multi-scale particle generation with screen blending and minimal fog (95th percentile, visibility $[10000,20000]$). *Clouds* employs Perlin noise with randomized opacity $\in [0.7,1.0]$, shadow intensity

$\in [0.2, 0.7]$, and blur scaling $\in [1.0, 3.0]$. *Raindrops* places 20-60 drops with radii $\in [3, 50]$ pixels and edge darkening $\in [0.4, 0.8]$.

**Noise and Resolution.** *Gaussian Noise* randomly applies either Gaussian ($\mathcal{N}(0, \sigma)$, $\sigma \in [0.05, 0.1]$) or salt-and-pepper noise (2-8% corruption, 50:50 ratio) with equal probability. *Defocus Blur* uses Gaussian kernels with size $k \in [3, 19]$. *Pixelation* downsamples by factors $s \in [2.0, 4.0]$ using bicubic interpolation for super-resolution tasks.

Parameter ranges for each degradation type are uniformly sampled within physically realistic bounds. See our codebase for the full implementation of this distortion library. Finally, the selected distortions are randomly ordered and sequentially applied to the clean image. Randomizing the application order reflects the non-commutative nature of compound distortions and further increases the diversity and realism of the visual outcomes. For all randomized sampling, we used a fixed seed of 42. This data augmentation process is summarized in Fig. 7.

Prompts describing distortions are auto-generated with GPT-4 (Hurst et al., 2024) to simulate the variability in natural-language queries likely to be provided at inference time. For each primitive distortion type $c$ in our vocabulary (e.g., haze, blur, noise), we query GPT-4 with "Provide 50 different ways a user might ask to remove $c$ from an image." This yields multiple phrasings per distortion (e.g., "remove haze," "dehaze this image," "clear the atmospheric fog," "fix the hazy artifacts"), promoting robustness to linguistic variation and paraphrasing.

To model realistic multi-artifact scenarios, we also generate compound prompts. For each pair or triplet of distortions $d = c_1, c_2, \ldots$ sampled during data construction, we query GPT-4 with "Provide 50 ways a user might ask to remove c1, c2, ... from an image." GPT-4 is required to mention all distortions in $d$ within a single natural instruction (e.g., "remove blur and color shift," "fix the blurriness and correct the color cast," "clean up both the motion blur and the hue distortion"). These compound prompts ensure that the model is trained on natural-language descriptions of *mixtures* rather than isolated categories.

All GPT-4 outputs were manually inspected to remove malformed or incomplete phrasings (e.g., missing distortion names, ambiguous requests). Our MDB dataset release includes our image pairs and natural-language prompts, as well as scripts used to generate and sample prompts.

To improve controllability, we further incorporate:

- **Partial prompts**: instructing the model to remove only a subset of degradations present in $I_{\text{dist}}$, requiring the model to learn selective restoration (e.g., input degraded with haze + rain + blur, prompt: "remove haze and blur").

- **Negative prompts**: instructing the model to remove degradations that are *absent*, which enforces that restoration actions are conditional on both image evidence and textual prompts. For instance, if the input is degraded with haze + blur and the prompt is "remove snow," the model should leave the image unchanged with respect to snow.

Approximately 20% of the training samples are partial prompts and 10% are negative prompts. The inclusion of partial and negative prompts is critical for teaching PRISM to respect expert instructions. Without them, the model tends to over-restore, indiscriminately removing all degradations it detects. By explicitly training on examples where the correct action is *not* to remove a present or absent degradation, PRISM learns to balance restoration fidelity with adherence to prompts.

The final training corpus consists of approximately 2.5M triplets, split $80/19/1$ across training, validation, and held-out test sets. For each clean image, multiple degraded variants and prompts are generated, increasing coverage of both degradation mixtures and linguistic variability. Fig. 8 demonstrates our dataset diversity, with examples of clean, distorted, and prompt triplets.

## C   EVALUATION

As discussed in Section 3, we evaluate PRISM along three complementary axes: (1) restoration under synthetic compound degradations, (2) downstream utility in real scientific datasets, and (3) zero-shot robustness to unseen real-world distortions. A summary of each evaluation testbed is provided in Table 6. Unless noted otherwise, all outputs are generated using a fixed random seed 42.

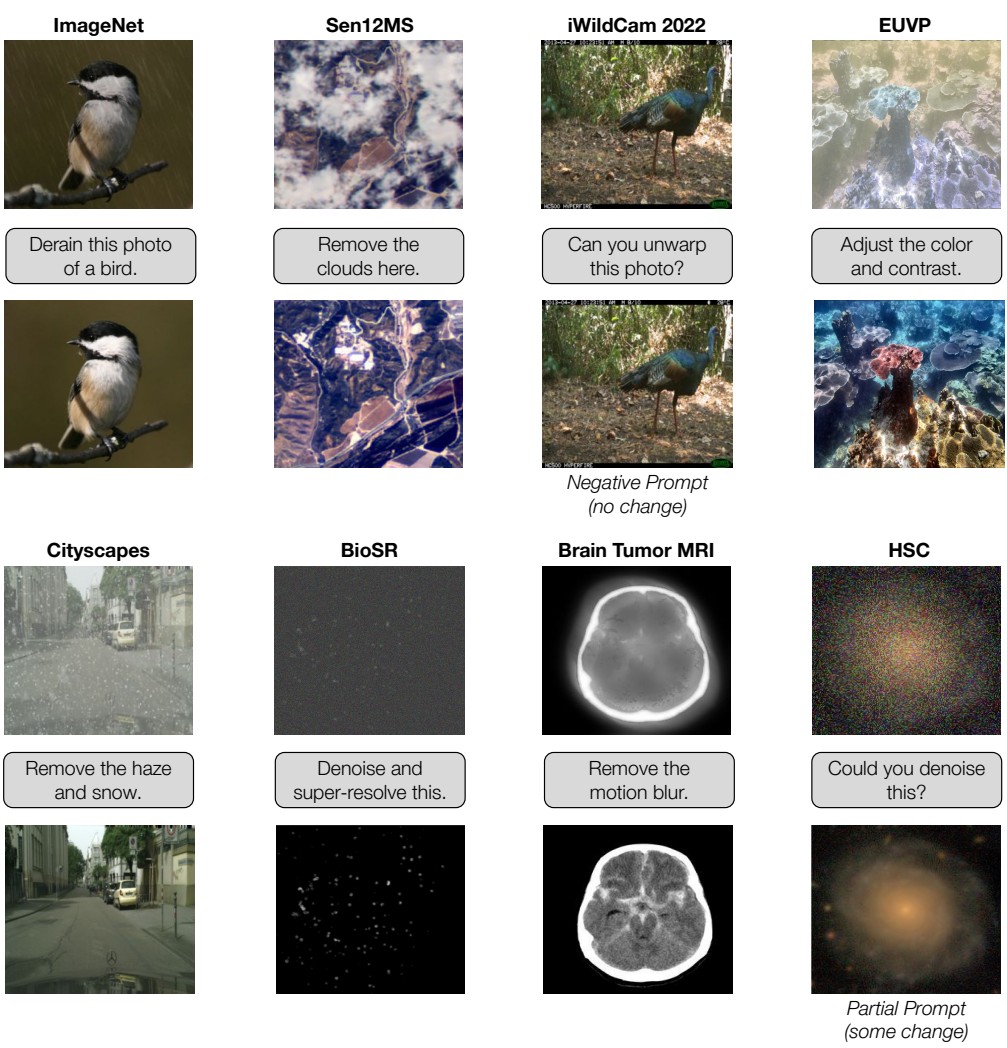

Figure 8: *MDB Examples.* Samples from our compound degradation dataset, with paired "clean" (top) and "distorted" (bottom) images, with corresponding prompts in between.

Table 6: We group evaluations into (1) synthetic compound degradations, (2) downstream utility in real scientific datasets, and (3) zero-shot robustness to unseen real-world distortions.

| Evaluation Setting | Task / Dataset | Dataset Size | Metric |
|---|---|---|---|
| **(1) Synthetic Compound Degradations** | | | |
| MDB | Synthetic mixtures from clean datasets | 25K | PSNR, SSIM, FID, LPIPS |
| **(3) Zero-Shot Robustness to Unseen Real-World Distortions** | | | |
| Underwater Imaging | UIEB (Li et al., 2019) | 890 | PSNR, SSIM, LPIPS |
| Under-Display Cameras | POLED (UDC) (Zhou et al., 2021) | 512 | PSNR, SSIM, LPIPS |
| Fluid Lensing | ThapaSet | 600 | PSNR, SSIM, LPIPS |
| **(2) Downstream Utility in Real Scientific Datasets** | | | |
| Land Cover Classification | Sen12MS-CR (cloudy Sentinel-2 patches) (Ebel et al., 2020; Schmitt et al., 2019) | 200 | Classification accuracy (ResNet50) |
| Wildlife Classification | iWildCam 2022 (camera traps) (Beuving et al., 2022) | 200 | Classification accuracy (Species-Net) |
| Microscopy Segmentation | BioSR (WF vs. SIM microscopy) (Gong et al., 2021) | 10K | Instance segmentation mIoU (MicroSAM) & Fluorescence quantification (MSE) |
| Urban Scene Understanding | Rooftop Cityscapes | 5K | Panoptic segmentation mIoU (RefineNet) |

**Downstream Utility.** Here, we provide specific details about how we constructed our novel benchmarking suite for evaluation over downstream utility. We re-purpose real datasets with distortions that present known challenges for models across remote sensing, wildlife monitoring, microscopy and weather, where ground truth labels are available *not* because the distorted images are annotated, but because undistorted views exist at different points in time or are collected from a more sophisticated sensor.

Rather than training task-specific models for each of these downstream tasks, we deliberately use off-the-shelf pretrained models. This design choice reflects a realistic scenario: domain experts are far more likely to apply widely available models for segmentation than to train bespoke models for each experimental setup. Using off-the-shelf models therefore provides a conservative estimate of restoration utility in practice and avoids confounding performance gains from joint training on dataset-specific distributions. If restoration improves the outputs of a generic model, this strongly suggests practical downstream utility beyond controlled benchmarks. In each of the four domains, we examine restoration performance on a specified distortion against the default set of automatically-detected distortions present in the input image. We do not compare against domain-specific restoration models (e.g., dedicated cloud removal networks) because our goal is to evaluate generalist models that can flexibly handle a wide variety of distortions; this broader applicability makes them more useful in scientific imaging, where degradations are diverse, overlapping, and often domain-shifted.

1. **Land Cover Classification:** To assess performance on land cover classification over satellite data, we select 200 cloudy satellite images degraded by cloud cover from the Sen12MS-CR dataset (Ebel et al., 2020), with land cover labels derived from temporally aligned, cloud-free views in the Sen12MS dataset (Schmitt et al., 2019). We evaluate using a ResNet50, trained on satellite imagery that includes minimal cloud cover (Papoutsis et al., 2023). It is important to evaluate on land cover classification because accurate identification of surface types (e.g., forests, croplands, urban areas) under degraded conditions like cloud cover directly underpins large-scale monitoring of climate change, biodiversity, and resource management.

2. **Wildlife Classification** Camera trap classification is critical for ecological monitoring, enabling large-scale biodiversity surveys without direct human observation. We evaluate our model on the task of species identification using *iWildCam 2022 Camera Trap* dataset (Beuving et al., 2022). Specifically, we use 200 nighttime wildlife images, after filtering out blank frames (no species) and sample frames with low confidence species predictions ($< 0.70$). Ground truth annotations are sourced from expert labels of alternate frames in the same camera trap sequence. Classification is evaluated using SpeciesNet (Gadot et al., 2024), a classifier trained on over 6 million camera trap images.

3. **Microscopy Segmentation** Next, we evaluate our model on the task of microscopy image segmentation, which informs the quantification of organelle morphology and dynamics, which are central to understanding cell physiology and disease. We build on the *BioSR dataset* (Gong et al., 2021) introduced by Qiao et al. This dataset was acquired using paired low resolution wide-field (WF, diffraction-limited) and super-resolved structured illumination microscopy (SIM) images of cellular structures (clathrin-coated pits) across a wide range of signal-to-noise ratios. In our setting, the WF images serve as noisy "distorted" inputs, while the corresponding high-SNR SIM sensor data provide the undistorted reference. This setup allows us to evaluate restoration not against simulated degradations, but against experimentally aligned ground truth. We measure performance by applying restored images to the downstream task of segmentation, using the microscopy foundation model MicroSAM model (Archit et al., 2025) to generate cell-structure masks, and report segmentation accuracy compared to the high-quality SIM annotations. This mirrors real-world use, where quantitative biological conclusions (e.g., about organelle morphology or cytoskeletal organization) depend critically on reliable segmentation.

4. **Urban Scene Understanding** We also evaluate our model on the task of cityscape scene understanding, which enables reliable monitoring of urban forests. To do so, we collected, labeled, and processed a novel *Rooftop Cityscapes* dataset for an additional setting: and haze in urban scenes. Specifically, we deployed fixed-position cameras on several building rooftops across multiple days under varying weather and lighting conditions. From each sequence, we manually identified and labeled frames with clear conditions to serve as the ground truth. We applied an off-the-shelf panoptic segmentation model pretrained on the original Cityscapes (Cordts et al., 2016) dataset to each distorted and restored frame. To ensure reliable comparison, we restricted evaluation to "stationary" classes (buildings, vegetation, and sky) while ignoring dynamic objects such as cars or pedestrians, which may change across frames and introduce label inconsistency. See Fig. 21 for qualitative examples from this custom dataset.

**Statistical Significance Evaluation** To assess whether Selective Restoration provided a statistically significant improvement over Full Restoration, we conducted paired hypothesis tests across repeated experimental runs. Each model was trained and evaluated with multiple random seeds on the same dataset splits (seeds 2, 42, and 420), yielding a distribution of results for each condition.

For every domain and downstream task, we computed paired differences between the two methods:

$$d_i = \text{Selective}_i - \text{Full}_i, \quad i = 1, \ldots, n$$

where $n$ is the number of runs (seeds/splits). This controls for variability due to dataset sampling and ensures that each comparison is made under identical conditions.

We then applied a two-tailed paired t-test to the differences $d_i$:

$$t = \frac{\bar{d}}{s_d/\sqrt{n}}, \quad s_d = \sqrt{\frac{1}{n-1}\sum_{i=1}^{n}(d_i - \bar{d})^2},$$

with $n-1$ degrees of freedom. The null hypothesis $H_0$ is that Selective and Full Restoration perform equally ($\mu_d = 0$). The p-value is computed as:

$$p = 2 \cdot P(T_{n-1} \geq |t|),$$

where $T_{n-1}$ is a Student's $t$ distribution with $n-1$ degrees of freedom.

1. If $p < 0.05$, we reject $H_0$ and conclude that Selective Restoration significantly outperforms Full Restoration.

2. If $p \geq 0.05$, we fail to reject $H_0$, indicating that the observed difference may be due to random variation.

## D  BASELINES

AirNet (Li et al., 2022a), Restormer (Zamir et al., 2022), and NAFNet (Chen et al., 2022a) represent strong encoder–decoder backbones widely used for low-level vision tasks, but they operate in an all-in-one setting without explicit modeling of compound effects. OneRestore (Guo et al., 2024) and PromptIR (Potlapalli et al., 2023b) extend this to multi-degradation scenarios: OneRestore introduces a one-to-composite mapping, while PromptIR conditions restoration on learned prompt embeddings. DiffPlugin (Liu et al., 2024) and MPerceiver (Ai et al., 2024) adopt modular or token-based conditioning, with DiffPlugin integrating contrastive prompt modules and MPerceiver encoding multiple degradation tokens. AutoDIR (Jiang et al., 2024) represents a task-routing approach, selecting subtasks adaptively during inference.

Among these, only PRISM employs a weighted contrastive loss to enforce compositional disentanglement in the embedding space. All other baselines use their standard supervision without this contrastive component.

All baseline models were re-trained or fine-tuned on the same set of primitive distortions as PRISM to ensure a fair comparison. This controls for training data bias and isolates differences in architecture and supervision. Following their original training protocol, all baselines (with the exception of OneRestore (Guo et al., 2019)) are trained on single distortions/primitives only, unlike PRISM which is trained on the full combinatorial mixutre set. For evaluation, we predefined a set of primitive degradations (e.g., blur, noise, haze, rain) and applied restoration pipelines consistently across models, so that all methods operated under identical inputs whether they were trained to remove distortoins independently or compositely. This avoids favoring baselines tailored to a specific distribution and provides a controlled setting for compound restoration.

Together, these baselines span backbone, prompt-driven, and diffusion-based strategies. Further details for fine-tuning and re-training our baselines and access to their implementations are included in the provided codebase linked above.

## E  ADDITIONAL ABLATIONS AND SENSITIVITY ANALYSIS

To better understand the contributions of individual design choices in PRISM, we conduct ablations on our held-out the Mixed Degradations Benchmark (MDB) test set.

**Semantic Content Preservation Module (SCPM).**  While latent diffusion excels at high-level denoising, it often loses fine-grained semantic or structural details critical in scientific imagery. To address this, we integrate the Semantic Content Preservation Module (SCPM), a lightweight decoder-side refinement block that adaptively fuses encoder and decoder features. Let $f_{\text{enc}}$ and $f_{\text{dec}}$ denote features from the encoder and decoder at a given UNet resolution. SCPM refines the decoder representation via

$$f_{\text{refined}} = \gamma(f_{\text{enc}}) \odot \text{Norm}(f_{\text{dec}}) + \beta(f_{\text{enc}}),$$

where $\gamma(\cdot)$ and $\beta(\cdot)$ are learned per-channel affine transformations (2-layer MLPs), and $\odot$ denotes element-wise multiplication.

The refined feature $f_{\text{refined}}$ is then passed through: (1) a residual convolutional block, (2) a lightweight self-attention layer, and (3) upsampled to the next scale. This design improves the preservation of edges, small textures, and domain-specific patterns, which are often blurred by diffusion bottlenecks. Appendix Figure 12 shows the architecture of the SCPM.

Quantitatively, removing SCPM drastically reduces MDB performance (see Table 7) and qualitatively, SCPM prevents content drift while recovering edges, textures, and small objects essential for downstream analysis (see Fig. 9).

**Effect of Temperature $\tau$.**  We sweep $\tau \in \{0.03, 0.07, 0.10, 0.20, 0.50\}$ while keeping all other hyperparameters fixed. For each $\tau$, we train the embedding module and freeze it before training the diffusion backbone. Figure 10 reports the average cosine similarity between degraded and clean

Table 7: *Ablation on PRISM's SCPM.* This refinement module improves compound restoration fidelity on MDB.

|  | PSNR ↑ | SSIM ↑ | LPIPS ↓ |
|---|---|---|---|
| After SCPM | **22.08** | **0.842** | **0.218** |
| Before SCPM | 16.65 | 0.654 | 0.566 |

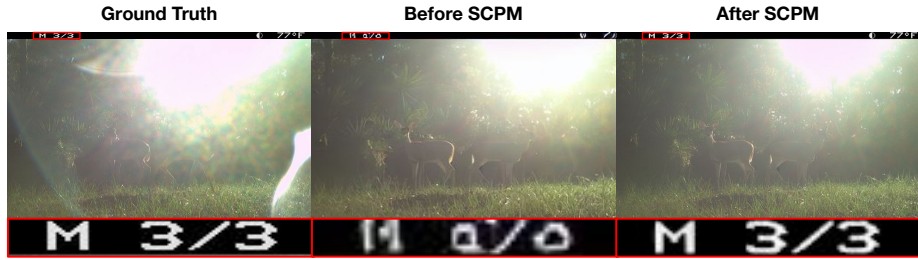

Figure 9: *Effect of SCPM on restoration fidelity.* Without SCPM (middle), restoration reduces degradations but alters scene details, leading to blurred text/textures and distorted object boundaries. With SCPM (right), fine structures are preserved, maintaining fidelity to the ground truth (left). By reintroducing encoder features at the decoding stage, SCPM retains spatial cues that are often lost in the bottleneck representation. This cross-scale fusion constrains the decoder to stay faithful to the input structure, reducing hallucinations and over-smoothing while preserving fine details critical for scientific fidelity.

views and the average difference between this positive similarity and the most confusable negative example (a large indicates better separation).

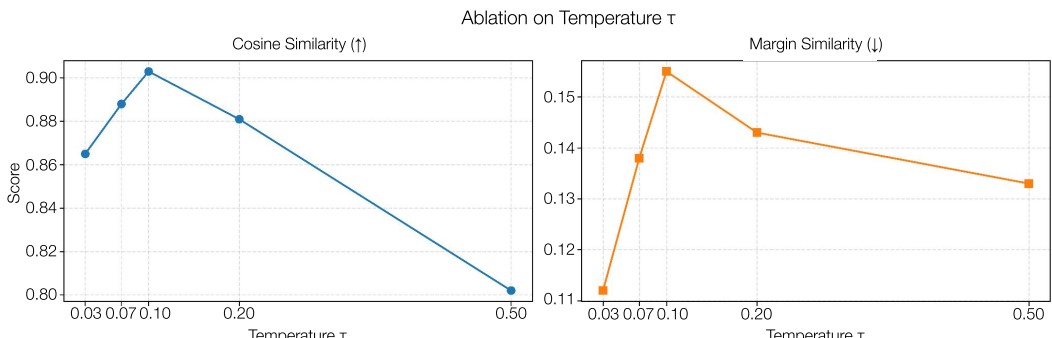

Figure 10: *Ablation on temperature $\tau$.* for the contrastive objective. We show the mean cosine similarity between degraded and clean views (left) and the mean gap between positive and hardest-negative cosine (right). Results are means $\pm$ standard error margin over 3 seeds.

We observe a sweet spot at $\tau \approx 0.1$, which maximizes separation. Very low temperatures ($\tau = 0.03$) over-emphasize hard negatives and reduce generalization; high temperatures ($\tau = 0.5$) soften negatives excessively, collapsing cluster structure and harming retrieval/accuracy. We therefore set $\tau = 0.1$ for all main results.

**Role of Jaccard Re-weighting.** Our setting involves *multi-label* supervision, where each example is associated with a set of labels rather than a single class. For the contrastive objective, we require a similarity measure that reflects the *degree of semantic overlap* between two such label sets. The Jaccard index is the canonical similarity measure for finite sets and is widely used in multi-label learning, retrieval, and segmentation (Lin et al., 2023; Menke et al., 2025)). Unlike cosine or dot-product similarity between binary label vectors, the Jaccard index ignores shared negatives, which is critical in sparse label spaces, normalizes by the size of the union, ensuring comparability across examples with different label cardinalities, and provides a bounded and interpretable similarity in

$[0, 1]$, yielding smooth interpolation between positives and negatives. These properties make Jaccard an ideal choice for weighting sample pairs in our supervised contrastive loss.

We compare our Jaccard-based similarity weighting against the following alternatives:

- **No re-weighting**: standard InfoNCE contrastive loss without contrastive reweighting,
- **Unweighted SupCon**: exact label matches treated as positives, all others as negatives,
- **Cosine similarity**: cosine between multi-hot label vectors,
- **Overlap coefficient**: $|y_i \cap y_j| / \min(|y_i|, |y_j|)$.

Each similarity score $s_{ij}$ is used in the same contrastive objective, with all training settings held fixed.

Table 8: *Effect of similarity function on downstream performance on the main dataset.* Values show absolute performance change relative to the Jaccard-based loss (ours).

| Similarity function | PSNR ↑ | SSIM ↑ | LPIPS ↓ |
|---|---|---|---|
| None | 19.52 | **0.734** | 0.154 |
| Unweighted SupCon | 20.63 | **0.799** | 0.388 |
| Cosine (labels) | 21.49 | 0.772 | 0.429 |
| Overlap coefficient | 21.35 | 0.784 | 0.332 |
| **Jaccard (ours)** | **22.08** | **0.842** | **0.218** |

Jaccard similarity yields the strongest performance. We suspect that alternatives such as cosine or the overlap coefficient systematically inflate similarity for high-cardinality label sets, reducing discriminativity. Unweighted SupCon performs worst because it discards all partial-overlap relationships, which are common in this dataset. Cosine similarity also correlates strongly with label count (Spearman $\rho = 0.42$), whereas Jaccard minimizes this dependency ($\rho = 0.07$), ensuring that similarity reflects the *fraction of shared labels* instead of the absolute number of labels present. Table 8 also shows that without any re-weighting in the contrastive loss (using a standard InfoNCE-based loss), overall performance decreases drastically.

Many sample pairs share only 1-2 labels. Jaccard naturally assigns intermediate similarity values in these cases, enabling "soft positives" that improve representation learning. We expect that the overlap coefficient overestimates similarity for such pairs, while Unweighted SupCon ignores them entirely.

**Role of Distortion Complexity**    Here we provide empirical evidence and theoretical motivation behind training on up to three distortions. To directly evaluate this design choice, we sweep the maximum number of distortions used during training over $N \in \{1, 2, 3, 4\}$ while holding all other factors constant (same set of clean images, same set of augmentations, same CLIP fine-tuning, same diffusion backbone). For each setting, we re-train PRISM from scratch and evaluate restoration fidelity (PSNR, SSSM, and LPIPS) on *images from MDB that are degraded by a single distortion*. We also evaluate training stability by the variance of the validation loss across epochs. Lastly, we retrain and evaluate the accuracy of a lightweight multi-label classifier on identifying the distortions present from the CLIP image encoder. Table 9 summarizes results across all distortion-count settings.

Table 9: *Impact of maximum distortion count ($N$) used in the training set.* Training on more than three simultaneous degradations reduces restoration fidelity, harms distortion classification accuracy, and destabilizes prompt-following behavior.

| N | Restoration Fidelity | | | Training | Classifier |
|---|---|---|---|---|---|
| | PSNR ↑ | SSIM ↑ | LPIPS ↓ | Stability ↓ | F1 Score ↑ |
| 1 | 24.35 | 0.942 | 0.112 | 0.014 | 0.91 |
| 2 | 20.65 | 0.923 | 0.126 | 0.018 | 0.88 |
| 3 | 18.73 | 0.842 | 0.218 | 0.022 | 0.87 |
| 4 | 16.98 | 0.741 | 0.401 | 0.047 | 0.61 |

From this sweep, we observe diminishing returns and eventual collapse beyond $N = 3$. Increasing $N$ from 1 to 3 expands the compositional coverage of the data augmentation pipeline, enabling the model to learn meaningful interactions among common distortions (e.g., blur+noise, haze+color shift). When 4 or 5 distortions are applied simultaneously, synthetic degradations become so severe that the underlying semantic content is ambiguously recoverable, hurting both restoration fidelity and

distortion-classification accuracy (F1 drops from 0.87 to 0.61). This aligns with the intuition that supervision becomes unreliable as compound degradations multiply. Thus, $N = 3$ maximizes the trade-off between representing realistic compound degradations (seen in underwater, microscopy, remote sensing, and camera-trap imagery) and maintaining well-posed, interpretable supervision for restoration.

**Role of Partial and Negative Prompts.**   Training with partial prompts (requesting removal of only a subset of degradations applied to an image) and negative prompts (explicitly requesting removal of degradations not present) enforces controllability. Without these cases, the model tends to over-restore, indiscriminately removing everything it detects. To evaluate this, we compute prompt faithfulness: for each prompt, we compare the predicted degradation labels before and after restoration against the degradations specified in the prompt. A restoration is counted as faithful if all requested degradations are removed while non-requested degradations are preserved. As shown in Table 10, including partial and negative prompts during training improves prompt faithfulness.

**Role of Prompt Diversity.**   We generate multiple natural-language variants for each distortion type (e.g., "remove haze," "clear atmospheric fog"). Limiting training to a fixed prompt format ("remove the effects of haze") only improves prompt faithfulness by 0.4%. Considering the tradeoff between accuracy and usability, we conclude that the benefits of linguistic variability outweigh this minor change in performance.

Table 10: *Effect of partial and negative prompts.* Including these improves prompt faithfulness (measured as proportion of outputs correctly following instructions).

| Training Setting | Prompt Faithfulness ↑ |
|---|---|
| w/o Partial or Negative Prompts | 61.4% |
| With Partial Prompts Only | 78.9% |
| With Negative Prompts Only | 85.9% |
| w/o Prompt Variation | 88.1% |
| With Partial + Negative Prompts and Prompt Variation (ours) | 87.7% |

**Sensitivity to Prompting Style.**   To clarify that PRISM does not require human labor at scale, we compare three prompting modes: (1) fixed manual prompts of the form "remove the effects of x, y, and z," (2) automated prompts produced by PRISM's multi-label distortion classifier, and (3) free-form user prompts paraphrased via GPT-4 (sampled from the pre-existing pool of variants of the fixed-prompt) to simulate real-world usage.

Table 11: Comparison of prompting modes on MDB. Automated restoration achieves 85–95% of fixed prompting performance.

| Method | PSNR ↑ | SSIM ↑ | LPIPS ↓ |
|---|---|---|---|
| Fixed Prompt | 22.08 | 0.842 | 0.218 |
| Automated Prompt | 20.84 | 0.824 | 0.229 |
| Free-form Prompt | 20.81 | 0.819 | 0.231 |

Automation provides near-manual performance while eliminating human specification entirely. This confirms that PRISM is not dependent on expert labor for large-scale deployment. Free-form prompting performs similarly, indicating that PRISM tolerates paraphrasing and user variation.

We evaluate PRISM's distortion classifier on the MDB test set spanning 14 primitive distortions and their synthetic combinations. The classifier achieves 0.94 accuracy on this benchmark, reliably identifying both individual degradations and multi-distortion mixtures. This high classification fidelity helps explain the strong automated restoration results reported in Table 11. When uncertain, the classifier tends to predict supersets of the true distortions (e.g., "blur + haze" instead of "haze"), a desirable bias in restoration where omission is riskier than mild over-specification.

However, MDB is a synthetic benchmark with cleanly defined distortion categories. It does not capture the open-ended, overlapping, and often ambiguous distortion mixtures found in real imagery. As a result, MDB accuracy cannot directly evaluate real-world generalization, and we expect classifier accuracy to be lower on real compositional degradations, where distortions are less separable and category boundaries are not well-defined.

**Sensitivity to Prompt Granularity and Accuracy** To explicitly quantify how different forms of prompt specification influence PRISM's behavior, we perform a controlled prompt-perturbation study on the MDB. For all experiments in this subsection, we use the same MDB test set validation images, ensuring that each prompt type is evaluated on *identical* inputs and degradation patterns.

For every image $I$ with ground-truth distortion set $\mathcal{D}$, we construct four prompt variants using GPT-4:

1. **Composite semantic:** A single combined instruction describing the same distortion set (e.g., "remove the effects of blur and haze"). These prompts are generated using a fixed template to avoid stylistic confounds. These prompts are generated using the form "remove the effects of x, y, and z."

2. **Over-specific:** Prompts augmented with redundant adjectives or intensity qualifiers (e.g., "remove mild Gaussian blur and strong chromatic haze"). For each set of distortions, we query GPT-4 with "Write 10 restoration instructions that tells a model to remove the following distortions x, y, and z. For each distortion, add one or two descriptive modifiers (e.g., mild, severe, colored, high-frequency, patchy)."

3. **Under-specified:** Prompts that target broad categories of restoration (e.g., true set: snow, prompt: "remove occlusions"). For each set of distortions, we query GPT-4 with "Write 10 broad, high-level restoration instructions that could apply to the following distortions x, y, and z. Use a generic category name such as 'occlusions' or 'artifacts.' Do not mention specific distortion types."

4. **Incorrect / negative:** Prompts specifying distortions *not* present in $\mathcal{D}$, sampled uniformly from the remaining distortion types (e.g., true set: blur+haze, prompt: "remove rain"). We used the fixed format "remove the effects of x, y, and z."

For each prompt variant, we encode the instruction using the frozen CLIP text encoder from our distortion-aware contrastive training stage, and condition the latent diffusion model with this embedding. Image restoration is run using the same sampling schedule and model weights across all conditions.

We report PSNR, LPIPS, prompt faithfulness, and the cosine similarity between the CLIP text embedding of the prompt and the CLIP image embedding of the degraded input (after applying the distortion-aware fine-tuning). The latter serves as a measure of *latent alignment* between the user instruction and the distortion evidence present in the image.

Table 12: Ablation on prompt granularity and accuracy. PRISM is robust to linguistic variations but sensitive to the correctness of the distortion set.

| Prompt Type | PSNR ↑ | LPIPS ↓ | Faithfulness ↑ | Alignment ↑ |
|---|---|---|---|---|
| Composite semantic | 22.08 | 0.218 | 0.98 | 0.93 |
| Over-specific | 22.01 | 0.225 | 0.96 | 0.89 |
| Under-specified | 20.30 | 0.241 | 0.78 | 0.70 |
| Incorrect / negative | 21.39 | 0.247 | 0.55 | 0.53 |

Results show that PRISM is invariant to changes in linguistic style (primitive-only vs. composite vs. over-specific), but is sensitive to whether the prompt specifies the correct *set of underlying distortions*. Importantly, even when prompts are incorrect or incomplete, the image-driven CLIP embedding pulls the joint embedding back toward an appropriate region of the latent space, limiting degradation. This supports the claim that PRISM is robust and image-conditioned, rather than overly dependent on prompt wording.

**Sensitivity to Local and Intensity-Aware Prompts.** We examine PRISM's behavior when given restoration instructions that reference either the strength (e.g., "slightly remove the haze") or the location (e.g., "remove haze from sky") of degradation removal. Although PRISM was not explicitly trained on scalar intensity levels or spatial masks, we observe that it can exhibit qualitative differences in response to intensity-related modifiers, with prompts such as "slightly" sometimes resulting in milder restorations (see Fig. 11). In contrast, spatially localized prompts are interpreted less consistently. This is unsurprising given that our data augmentation pipeline introduces degradations uniformly across the entire image, meaning the model is primarily exposed to globally applied

distortions during training. Future work incorporating localized augmentations and corresponding context-specific prompts may help improve performance in this setting.

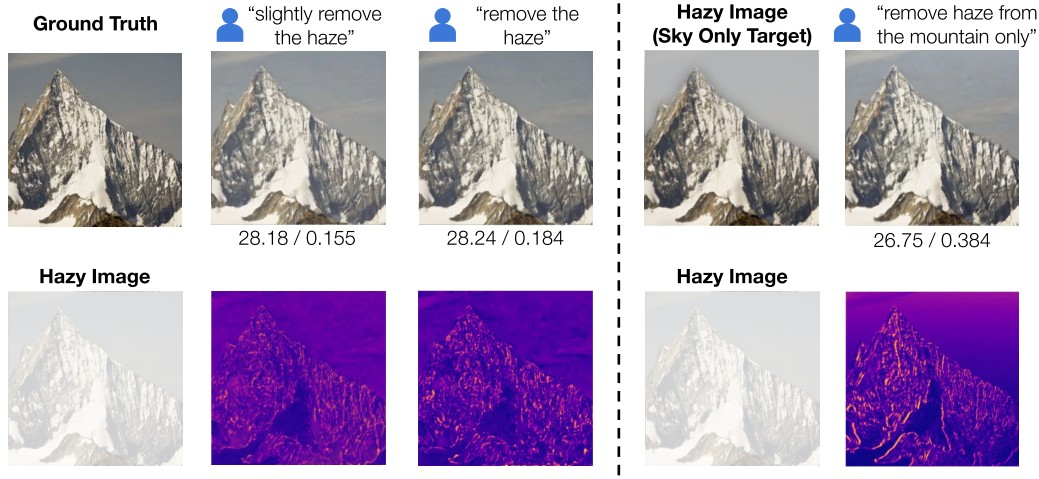

Figure 11: *Responses to intensity- and location-aware prompts.* PRISM shows qualitative variability with different prompt phrasings: milder restoration tends to occur under "slightly remove the haze," while stronger dehazing appears under "remove the haze." Spatial specificity remains limited, and the model generally applies restoration globally even when local changes are requested (e.g., "remove haze from sky"), likely reflecting its training on global augmentations. Bottom row: heatmaps visualize pixel-wise error, and PSNR/LPIPS values are reported below.

**Cost and Latency Analysis** In this section, we benchmark efficiency, memory usage, and latency under standardized conditions to evaluate the practical deployability of PRISM relative to both lightweight restoration models and state-of-the-art diffusion-based baselines.

All evaluations were conducted on a single NVIDIA A100 (40GB) GPU using a fixed input resolution of $256 \times 256$. We report FLOPs (via the 'thop' library), peak GPU memory usage (via PyTorch profiling tools), and average per-image latency (ms). All models were evaluated using identical batch size, mixed-precision (fp16), and PyTorch's benchmarking mode for consistent measurement.

Table 13: *Efficiency comparison under standardized conditions.* Despite added controllability and SCPM modules, PRISM matches the runtime and memory footprint of strong diffusion-based baselines.

| Method | FLOPs (G) | Memory (GB) | Latency (s ↓) |
|---|---|---|---|
| AirNet (Li et al., 2022a) | 46 | 2.1 | 0.3 |
| Restormer$_A$ (Zamir et al., 2022) | 118 | 4.6 | 0.5 |
| NAFNet$_A$ (Chen et al., 2022a) | 104 | 4.2 | 0.5 |
| OneRestore (Guo et al., 2024) | 136 | 5.8 | 0.6 |
| PromptIR (Potlapalli et al., 2023b) | 128 | 5.4 | 0.6 |
| DiffPlugin (Liu et al., 2024) | 145 | 6.5 | 2.63 |
| MPerceiver (Ai et al., 2024) | 132 | 5.9 | 1.98 |
| AutoDIR (Jiang et al., 2024) | 138 | 6.0 | 2.24 |
| **PRISM (manual)** | 141 | 6.1 | 2.55 |
| **PRISM (automated)** | 141 | 6.1 | 2.87 |

As expected, lightweight encoder–decoder models such as AirNet offer the fastest throughput and lowest memory usage, but struggle with generalization and complex distortions as shown in the main text of the paper. Transformer-based or prompt-conditioned models (Restormer, NAFNet, PromptIR) increase compute due to deeper backbones and multi-branch design.

Among diffusion-based models, PRISM operates at a similar computational cost to leading baselines (DiffPlugin, AutoDIR, MPerceiver), with a modest increase in FLOPs and memory due to our SCPM conditioning module. Importantly, this added capability yields improved downstream fidelity (Tables

2, 5, and 9 in main text and Appendix). The distortion classifier used in automated restoration incurs minimal additional cost compared to manual restoration. We believe this demonstrates that prompt-guided, compound-aware restoration is achievable without compromising deployability—making PRISM well-suited for integration into scientific workflows that require interpretability and modularity alongside high-quality results.

# F  ADDITIONAL FIGURES

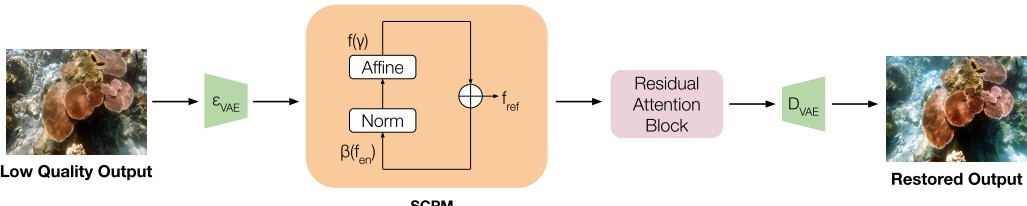

Figure 12: *Semantic Content Preservation Module (SCPM).* Encoder features $f_{\text{enc}}$ are used to generate adaptive affine parameters $\gamma(f_{\text{enc}})$ and $\beta(f_{\text{enc}})$, which modulate normalized decoder features $\text{Norm}(f_{\text{dec}})$. The refined features $f_{\text{refined}}$ are then processed by residual and attention blocks before final decoding by $D_{\text{VAE}}$. This adaptive fusion preserves fine structures such as edges, textures, and small objects that are critical for scientific imaging tasks.

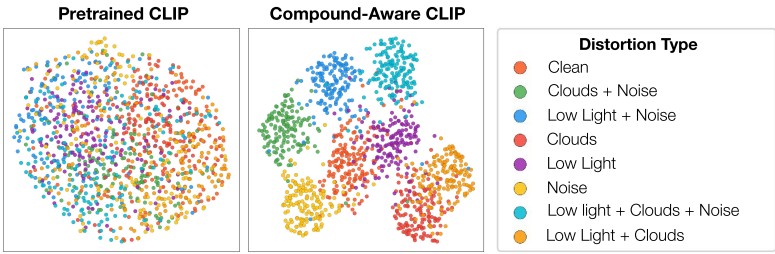

Figure 13: *Contrastive disentanglement of distortion embeddings.* UMAP projections of $f(I_{\text{dist}})$ from 10K samples in the MDB, across a subset of distortion classes. Left: CLIP entangles distortions with semantics. Right: Our weighted contrastive loss achieves clear separation while aligning compounds with their primitives. Overall, without contrastive disentanglement, embeddings of compound degradations collapse into unrelated regions, forcing the model to treat them as unseen categories. This can lead to artifacts or overcorrection.

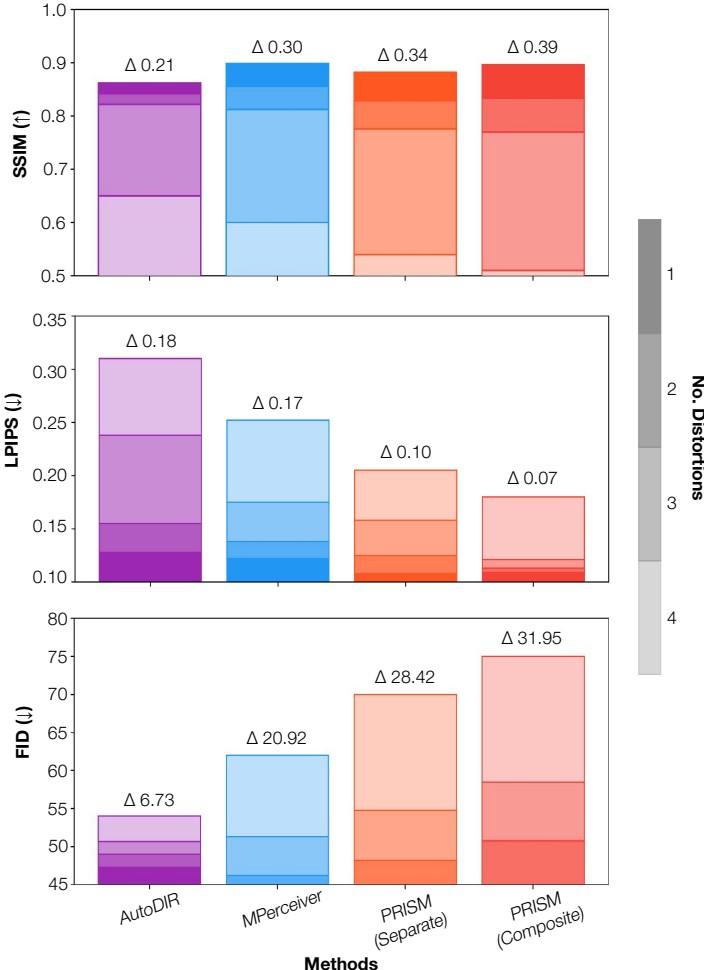

Figure 14: *PRISM trained on composite examples scales best with the number of distortions.* This outperforms PRISM trained on each degradation separately as well as comparable baselines (MPerceiver and AutoDIR), emphasized by the Δ (change in performance across test images with 1 vs. 4 distortions) above each bar.

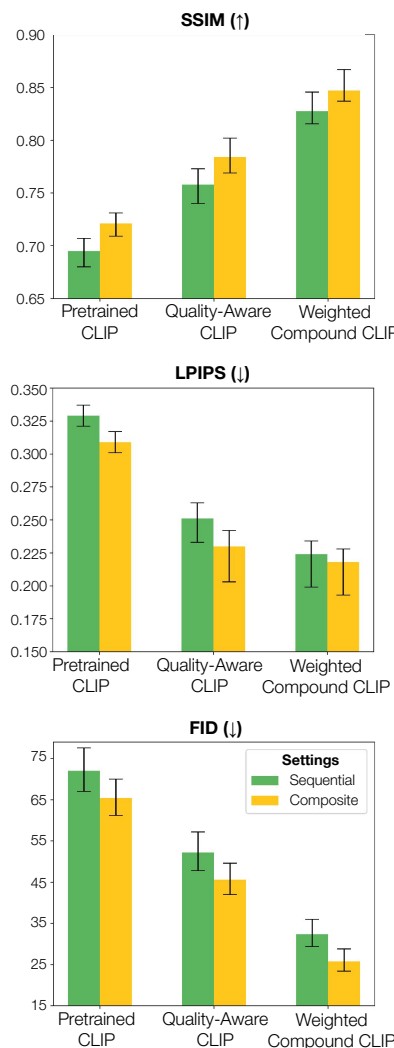

Figure 15: *Latent disentanglement of distortion types enables faithful stepwise and single-shot restoration.* The contrastive loss closes the gap between sequential and composite prompting.

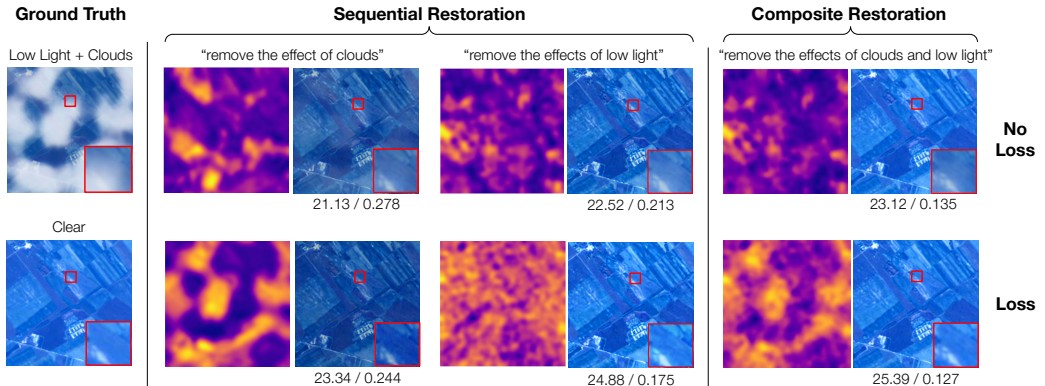

Figure 16: *Contrastive disentanglement of distortions helps separate distortions from each other and from semantic content, enabling higher-fidelity sequential and composite restoration.* Cross-attention maps (left of each output) show how the model attends to distortions. Without PRISM's contrastive disentanglement (top), sequential restoration preserves artifacts and fails to isolate degradations. With the loss (bottom), embeddings cleanly separate distortions (e.g., clouds vs. low light). This separation not only prevents distortion types from interfering with one another, improving sequential restoration by reducing error accumulation, but also enables the model to accurately target and remove multiple degradations simultaneously, as seen in the composite restoration outputs. We report PSNR/LPIPS metric values below each output.

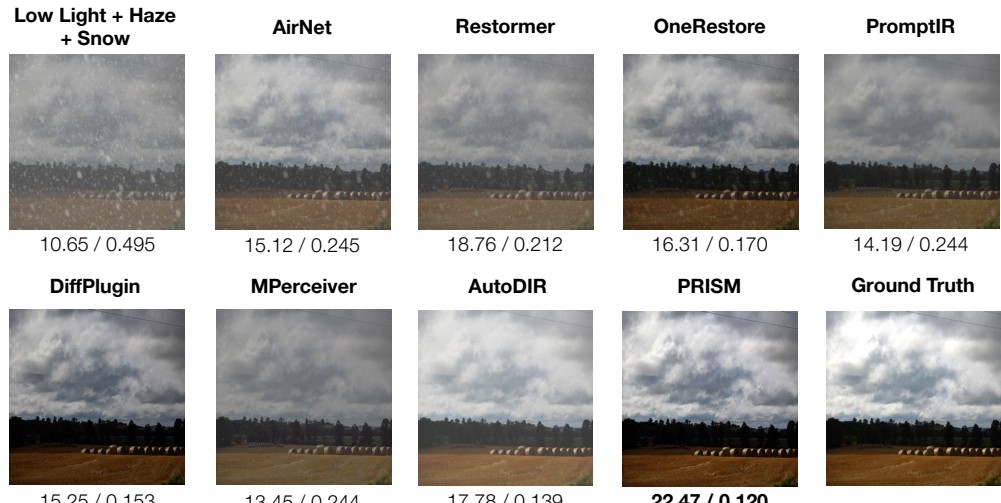

Figure 17: *Qualitative outputs on the Mixed Degradations Benchmark (MDB).* Example of a low-light + haze + snow composite evaluated across baselines. We report (PSNR/LPIPS) below, with the best results in **bold**. While prior methods reduce some degradations, they leave residual haze (AirNet, PromptIR), oversmooth texture (Restormer, MPerceiver), or introduce artifacts from over-correction (OneRestore, AutoDIR, DiffPlugin). PRISM produces the most faithful reconstruction, recovering both sky and foreground with minimal artifacts, closely matching the ground truth. This illustrates the strength of compositional latent disentanglement: PRISM not only removes multiple degradations simultaneously but also resists the tendency to over-restore, yielding outputs that are both high fidelity and scientifically faithful.

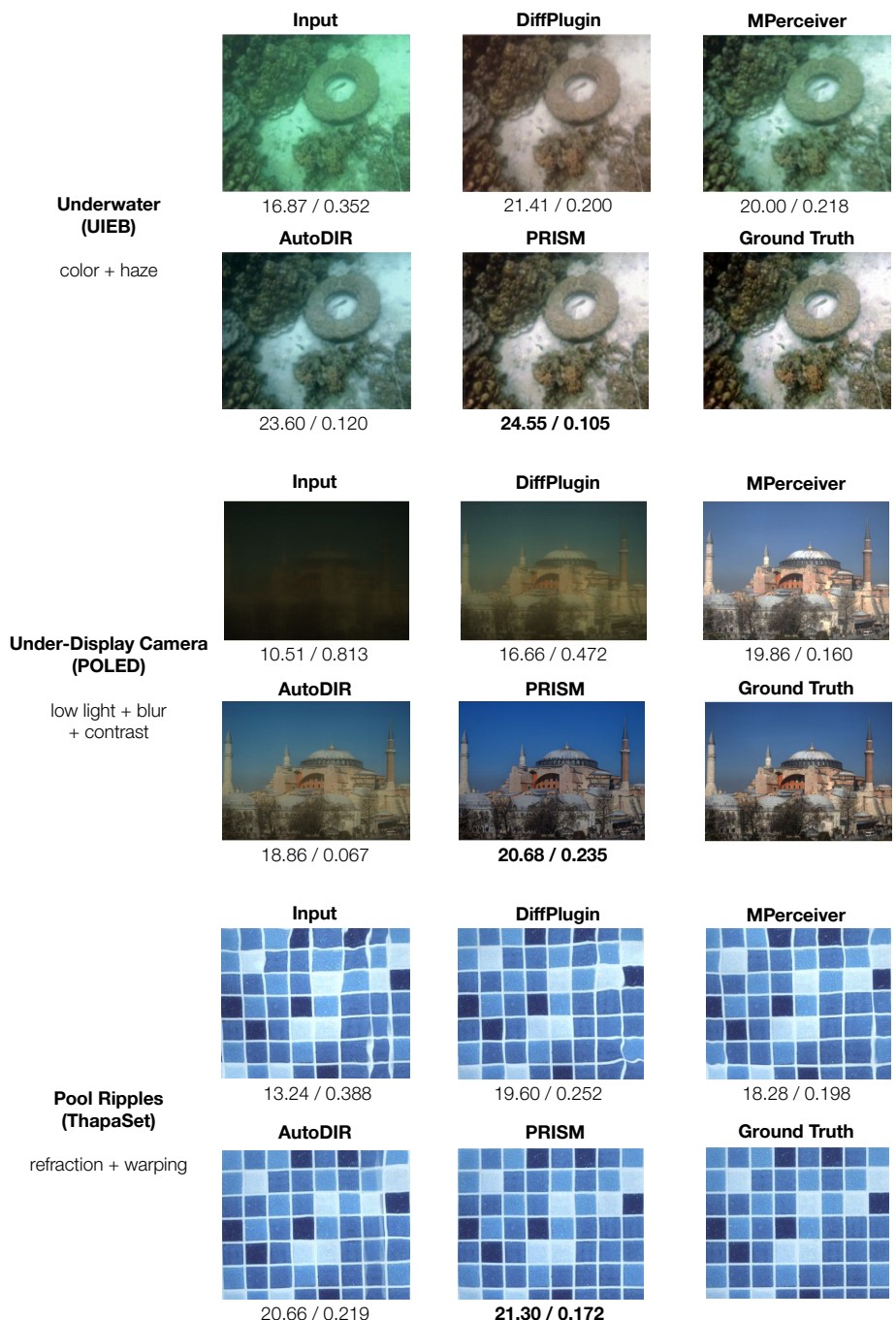

Figure 18: *PRISM best removes unseen compound degradations.* PRISM restores real-world images with degradations outside its training set in underwater imagery, under-display camera images, and fluid lensing. In all cases, it produces faithful restorations that most closely match the ground truth, showing strong single-shot generalization compared to similar diffusion baselines. We report PSNR/LPIPS metric values below, with the best results in **bold**.

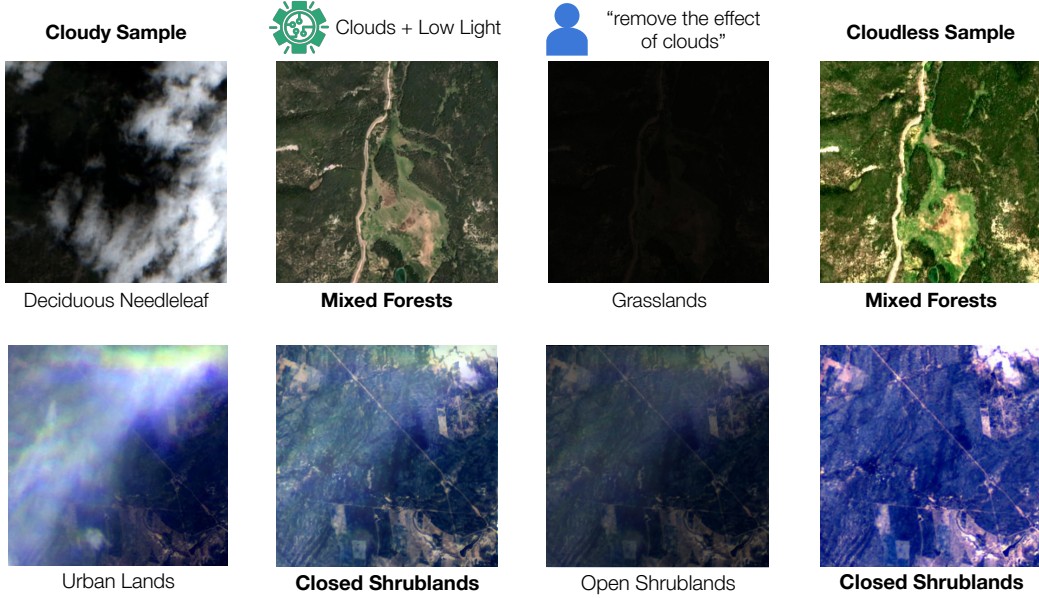

Figure 19: *Remote sensing classification under cloud occlusions requires full composite restoration.* In this Sentinel-2 example, removing only clouds (middle-left) reveals incomplete information and leads to a misclassification. Full composite restoration (middle-right), correcting both clouds and low light, recovers the underlying landscape with high fidelity and matches the ground-truth class, in **bold**.

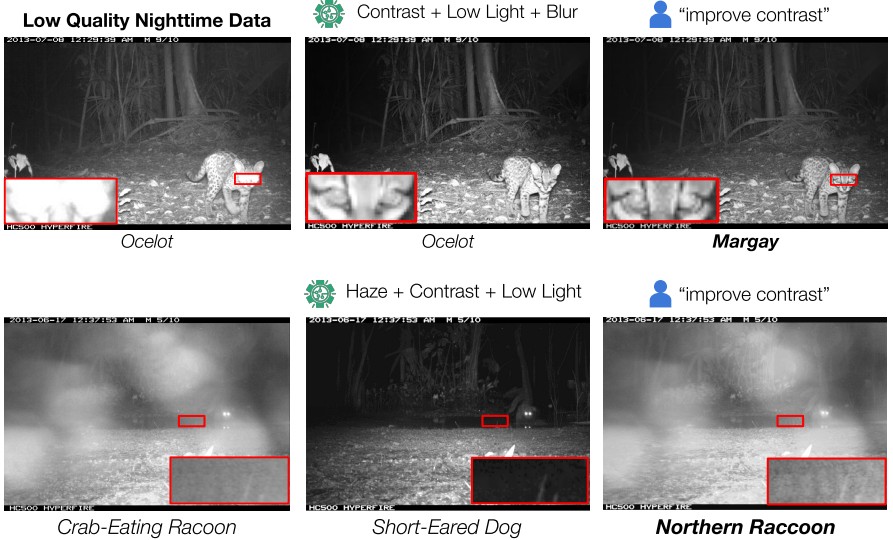

Figure 20: *Selective restoration helps with camera trap classification under compound nighttime degradations.* On the Rooftop Cityscapes dataset, frames suffer from haze and low-light conditions. Only improving contrast aids recognition of nocturnal species, while over-restoration (e.g., removing haze) can alter image content, obscure subtle texture cues, or introduce artifacts that mislead classification—sometimes even changing the perceived species. We **bold** the classification outputs that matches expert-provided labels.

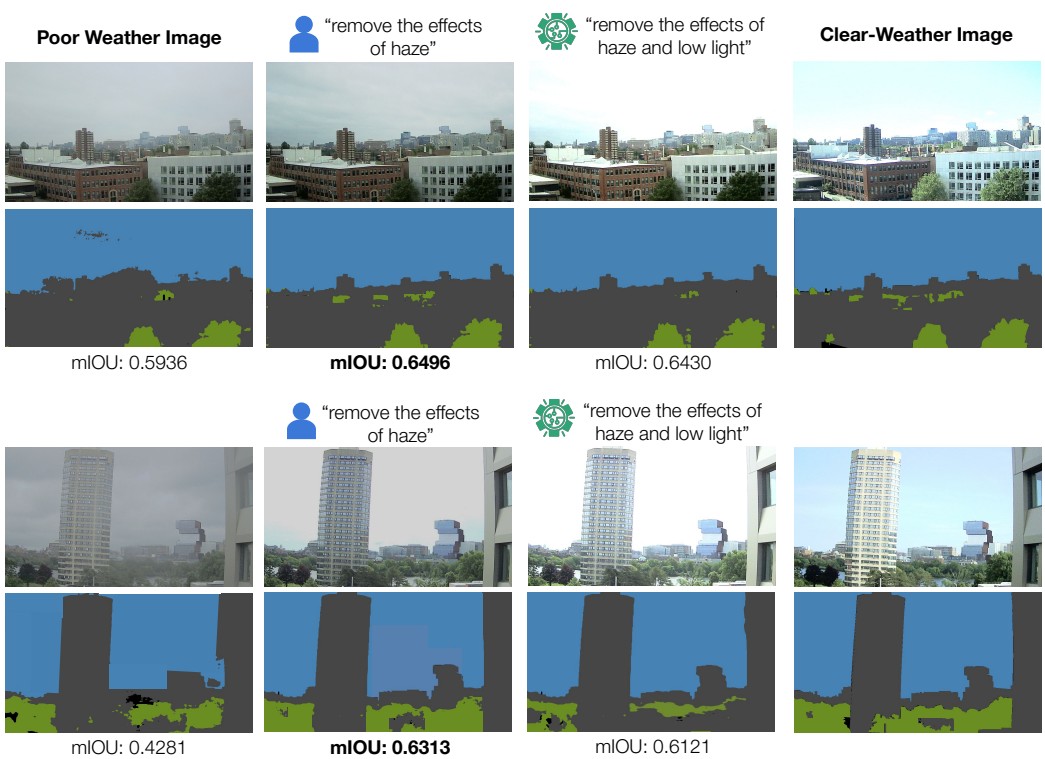

Figure 21: *Selective restoration helps with urban scene understanding under haze and low light.* Rooftop Cityscapes examples show how selective restoration affects scene understanding. Removing haze alone improves mIoU, while attempting to also remove low light leads to over-correction and lower segmentation accuracy.

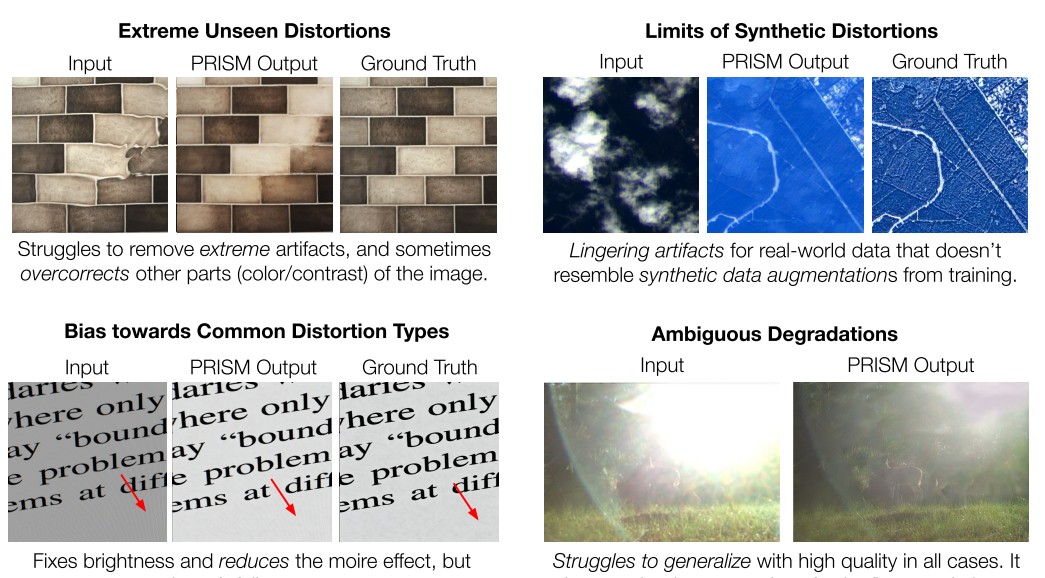

Figure 22: *Failure modes of PRISM on challenging degradations.* Top-left: Extreme unseen distortions cause incomplete restoration and overcorrection of color/contrast. Top-right: Overfitting to synthetic distortions leaves lingering artifacts when applied to real data that diverges from training augmentations. Bottom-left: Overfitting to common distortions partially reduces moire but fails to fully restore fine details. Bottom-right: Ambiguous degradations (e.g., solar flares, glare) remain difficult to generalize without explicit training examples.

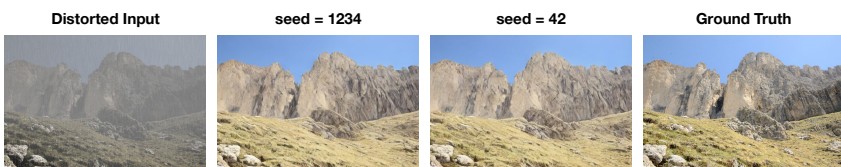

Figure 23: *Qualitative impact of random seed and stochasticity on restoration outcomes.* Different seeds produce slightly varied outputs, reflecting both diffusion sampling variability and embedding initialization. While global structure remains stable, fine details may differ, underscoring the importance of evaluating consistency across multiple runs.