# OpenReview forum: "Seeing Through the PRISM: Compound & Controllable Restoration of Scientific Images"
_ICLR.cc/2026/Conference — ICLR 2026 Poster_

### Official Review · Reviewer_dhPx · 2025-10-21

**Soundness:** 1
**Presentation:** 3
**Contribution:** 2
**Rating:** 2
**Confidence:** 3

**Summary:**

The authors argue that the current unified processing pipeline adopted for image restoration in the scientific field may introduce redundant restoration types, which could counter productively lead to the loss of authentic information. Therefore, the authors aim to provide a controllable mechanism, Controllable Diffusion,  to enable experts to select specific restoration types to avoid the aforementioned dilemma.

**Strengths:**

A.The authors present a novel perspective in addressing the issue.

B. The amount of work undertaken is solid.
>- They constructed a brand-new dataset and employed the latest relevant methods (mostly works from 2024) to demonstrate the superiority of the proposed approach.

**Weaknesses:**

I think subjective assumptions and the unconvincing problem-solving approach are the most prominent issues.

A.Subjective assumptions
>-  A.1 Taking underwater image restoration, as mentioned in the article, as an example. The authors identify three types of distortions: low light, scattering, and wavelength dependency. Since these distortions act on the same image, any single restoration method may inadvertently damage useful information. Therefore, the core of fidelity preservation should lie in the precision of each restoration. However, the authors' equivalence of precision with controllability over the quantity and types of restoration is a unconvincing subjective assumption.

>- A.2 It is possible that the most appropriate combination of restoration types has already been implicitly learned by the model through training. However, the authors have neither experimentally nor theoretically refuted the aforementioned viewpoint nor provided substantive support for their own hypothesis.

>- A.3 The authors arbitrarily set the number of distortion types to 3, which is overly simplistic and lacks justification. This approach may fails to flexibly handle complex scenarios.

B.Unconvincing problem-solving approach
>- B.1 The core idea of artificial intelligence is to employ machine intelligence to replace human thinking. When encountering problems, the focus should be on solving them directly, rather than reversely introducing human labor costs to empower machines.

>- B.2 Although the experimental results show some improvement, these enhancements are not significant. Moreover, there is a lack of relevant analysis regarding the extent to which such improvements stem from the investment in human labor costs and whether they offer cost-effectiveness. Additionally, the ablation experiments fail to fully elucidate the impact of distortion type settings and their dependence on accurate expert guidance.

>- B.3 Assuming your hypothesis is correct, why not just solve the problem in a software engineering way? Let a single model specialize in removing a single type of noise,  we design control codes to allow experts to manually execute several actions to process the images. Therefore, what is the point of incorporating NLP models for processing opinions of experts and coupling the problem together to train a model?

**Questions:**

The same as the above analysis in Weaknesses.

---

> ### Author Response · Authors · 2025-11-20
> **Addressing Part A (Subjective Assumptions)**
>
> We thank the reviewer for the detailed feedback, especially from the angle centered around the assumptions and design decisions that define our method. This perspective is valuable, and we respond to each concern below to the best of our ability; please let us know if there are any further comments, concerns, or misunderstandings. First, we address the comments about the subjective assumptions made in this work.
>
> * **A.1:** This is an insightful point! We would like to clarify that we do not equate "precision" with "controllability." Precision in PRISM refers to fidelity-preserving recovery of scientifically relevant signals. Controllability enables precision by preventing over-restoration (i.e., removing valuable subtle features such as fine ecological cues or weak microscopy structures). Automatic pipelines often overcorrect, while expert-specified prompts help preserve what matters. We **emphasize this in Sec. 2.1** and with Fig. 6 Table 4.
> * **A.2:** In Section 4.2, we do not necessarily argue that PRISM implicitly learns the ideal combination of primitives for generalization. State-of-the-art all-in-one baselines, many of which attempt to implicitly learn combinations, perform substantially worse in complex mixtures (Table 1) and in zero-shot settings (Table 2). Even diffusion-based generalists (MPerceiver, AutoDIR) struggle when distortions overlap nonlinearly or as complexity grows (Fig. 3). Without explicit structure, models tend to collapse mixtures into entangled latent spaces and overcorrect in novel scenarios. We **revise the language in 4.2** to qualify these claims.
> * **A.3:** We chose 3-way mixtures to balance difficulty, realism, and interpretability. Based on empirical experiments from previous iterations of PRISM, using 4+ degradations leads to severe signal collapse (**Appendix E, under "Role of Distortion Count in Data Augmentation Pipeline"**). Moreover, considering the real-world data evaluated on in Sections 4.2 and 4.3, most real-world examples contain 3 or fewer active degradations, and compositional prompts beyond this become less practical and interpretable.. Prior works tackling compound image restoration (e.g., OneRestore, AllRestorer) make similar design choices.

---

> > ### Author Response · Authors · 2025-11-20
> > **Addressing Part B (Approach)**
> >
> > Here, we address the comments on the approach itself.
> >
> > * **B.1:** We respectfully disagree with the premise that "AI should always replace human thinking," and argue instead that "AI should augment human thinking." As argued in Section 4.2 and lines 67-69 of the paper, in scientific domains, full automation can decrease reliability. Experts need control, not replacement, to better suit restoration to the nuances of their data. However, we want to clarify that PRISM makes restoration more automated than most existing pipelines for sequential restoration and manual parameter tuning. In addition to enabling automated restoration (now **clarified in Section 3.3**), in the interactive setting, the expert only provides high-level compound restoration intent ("remove the effects of haze and blur") without having to specify parameters or do manually editing in multi-step pipelines.
> > * **B.2:** While standard metric gains are modest in aggregate, they are substantial in complex scenarios. As shown in Fig. 3 and Table 2, our method consistently outperforms all baselines on complex, mixed-degradation cases. We do sometimes underperform on simple, single-distortion inputs, but this is expected, as most baselines are explicitly trained for this task. We note that the modest gains in Table 3 are rooted in the fact that not all images require restoration to be useful for a downstream task, especially when the downstream model is already robust to low data quality. The margins in Table 3 reflect this: in easier scenarios, perfect restoration offers diminishing returns. To the reviewer's point about expert effort, we explicitly discuss the tradeoff between human labor and automation in our response to point B.1. At a high level, **our ablations in Appendix E (Tables 7-9)** show that prompt diversity, partial restoration supervision, and negative prompting during training support reasonable prompt faithfulness and robustness to linguistic variations. **Table 8 shows that this diversification in the language of prompts has minimal impact** on restoration fidelity, as long as the expert is not ambiguous with their langauge. We include **quantitative analysis in Table 10 measuring restoration performance across prompt variants** (see Appendix E, under "Robustness to Natural Language Prompt Variation"). Results show high consistency in PSNR, SSIM, and LPIPS across all variants (<0.5 dB PSNR variation), indicating strong robustness to linguistic variability. Our ablation on including partial and negative prompts during training **(Appendix E, Table 9)** shows that these improve prompt faithfulness by over 6%, helping PRISM better follow complex, "incorrect," or selective instructions. **Fig. 10 (see Appendix E, under "Response to Local and Intensity-Aware Prompts")** shows qualitative examples showing varied levels of restoration under varied severity levels.
> > * **B.3:** The reviewer's point about the overall approach is well-taken. While we agree with the perspective that the simplest, most direct solutions are often better, a modular "one-model-per-distortion" pipeline fails in several fundamental ways. First, we discuss in Sections 1 and 4.1 and Appendix F Figure 16 how sequential restoration (removing each artifact one-at-a-time) may introduce errors that subsequent stages amplify. This motivates our emphasis on tackling composite restoration in a simultaneous manner. Additionally, while specialized models cannot handle unseen compounds, PRISM's scale and training pipeline handles these zero-shot (Table 2) and in an automated fashion. With a "software engineering" approach, experts would need to manually pick the pipeline order, placing more of a burden on human effort. Lastly, incorporating natural language prompts offer flexible, scalable control beyond what fixed pipelines are capable of.
> >
> > We hope these clarifications address the reviewer’s concerns and more clearly highlight the novelty and necessity of PRISM.

---

> > > ### Comment · Reviewer_dhPx · 2025-11-25
> > >
> > > ***B1*** What I really meant is that artificial intelligence should liberate human labor costs, allowing humans to focus on more creative fields. Combining restoration types clearly doesn't fall within the realm of human creative thinking. Theoretically, the act of combining restoration types should be replaced by machines. If there are 10,000 images, experts need to conduct 10,000 analyses and write 10,000 suggestions (the purpose of training machine learning models is to apply them to data scales far larger than the training set). Machine learning is precisely designed to handle large-scale processing. When dealing with a small number of images, the focus is on whether the machine learning method follows objective laws for restoration.
> > >
> > > ***B2*** Considering that the experimental results haven't shown significant improvements, we should focus on the problem-solving approach or theoretical contributions when evaluating the value of innovation. The concerns regarding the problem-solving approach have been addressed in ***B1***.
> > >
> > > ***B3*** The composite restoration problem proposed by the authors has resolved some of my concerns. Manually selecting restoration operations to perform might just involve clicking a few buttons. In contrast, natural language prompts mean that clicking actions are replaced by typing, which theoretically involves higher labor costs. Meanwhile, the flexibility brought by natural language prompts implies an increase in uncontrollability.

---

> ### Comment · Reviewer_dhPx · 2025-11-25
> **Response to the Addressing Part A (Subjective Assumptions)**
>
> Thank you very much for the author's responses. I still have some questions, and I'll raise my score if the author can solve my problems.
>
> >- ***A1*** Has there been any specific testing on the impact of the granularity and accuracy of prompts on restoration? Are the prompt features generated by the language model after analyzing expert inputs visualized and explained? Conducting an ablation study by directly changing prompt features is crucial for the scientific validity and understanding of the method.
>
> >- ***A2*** My question isn't merely about whether PRISM implicitly learns the ideal combination. Instead, an ideal advanced model should learn decision-making equivalent to the expert prompts you've proposed. I acknowledge the entanglement issue you mentioned, so transforming this task into a two-stage process, namely combination decision-making + restoration, is a reasonable idea. However, what you've done is to replace the decision-making process with human labor, which is hardly a solution recognized by machine learning conferences.
>
> >- ***A3*** My understanding is that the number 3 is just derived from experimental experience and lacks reliability.

---

> ### Author Response · Authors · 2025-11-27
>
> We sincerely thank the reviewer for the quick response and the critical thought put into evaluating our work, and apologize for any misunderstanding on our part. We found the follow-up comments constructive and have conducted additional analyses and ablations to address them directly. Any updates to the paper and supplementary materials are highlighted with blue text.
>
> * **A1:** Thank you for raising this. We fully agree that more rigorously studying the sensitivity of PRISM to prompt structure is important. We previously stated that our ablations in Appendix E Table 10 show that prompt diversity, partial restoration supervision, and negative prompting during training support reasonable prompt faithfulness and robustness to linguistic variations. However, the idea of explicitly perturbing prompt features is well-taken. We have now added a new ablation study that varies both granularity and accuracy of prompts, quantifying the model’s robustness and interpretability (Appendix E, under "Sensitivity to Prompt Granularity and Accuracy"). PRISM is robust to linguistic variance but sensitive to correctness of the **set of distortions**, not the exact wording. Even under mis-specified prompts, PRISM tends to follow the distortion set in the *image embedding*, suggesting that **image-driven guidance strongly modulates the text embedding**, reinforcing your point about implicit decision-making. We believe that this strengthens the scientific validity of our method and directly addresses your point on controlled prompt feature ablations.
> * **A2:** We agree fully that an advanced model should *not* rely on human decision-making at scale. PRISM was designed to support both expert input **and** automation, and we regret that the paper did not sufficiently emphasize the automated pathway. PRISM already **uses the CLIP image encoder as a multi-label distortion classifier**, enabling *automatic, prompt-free restoration*. This is described in Sec. 3.3, but we have modified our contributions list and the language in Sec. 4.2 to better emphasize this. In Appendix E, under "Sensitivity to Prompting Style", we include new experiments comparing restoration performance with fixed prompts ("remove the effects of x"), automated restoration, and free-form prompts (natural language variants of the same instruction). Our results show that PRISM is *not* significantly dependent on human decision-making. The system already includes an automated pipeline; expert-in-the-loop prompting is an optional override for scientific contexts where incorrect restoration can erase faint signals. However, performance *is* dependent on how "correct" or "complete" the input prompt is.
> * **A3:** This is a reasonable concern. As stated in our previous response, the restriction to training on up to three distortions is *not* arbitrary but chosen to balance between (1) preserving the underlying semantic signal and (2) real-world mixture complexity. We've expanded our experiments in Appendix E under "Role of Distortion Complexity" beyond evaluating training with 1-4 distortions on PSNR/SSIM/LPIPS with single-distortion removal to training stability and embedding quality. Our results show that training on beyond 3 distortions, performance **degrades** due to exponential growth of mixture classes, diminishing marginal benefit, reduced prompt comprehensibility, and CLIP-text ambiguity (“remove blur, haze, rain, low light, compression” becomes ill-defined). We explicitly state in this supplementary discussion that 3 reflects the observed natural joint-degradation structure in scientific imagery (underwater, microscopy, camera traps, satellite), where mixtures rarely exceed 2–3 overlapping degradations, as shown in domain literature and our real-world datasets (UIEB, POLED, ThapaSet in Section 4.2; Figure 5; and our domain-specific datasets from Sec. 4.3). In particular, we point to Fig. 6 in Sec. 4.3.1 (now 4.3) and Figs. 19-21 in Appendix F, where we show qualitative examples of automated restoration, compared to outputs with manual prompts. We believe that this addresses the reviewer’s concern by grounding the choice in both empirical analysis and real-world imaging statistics.

---

> > ### Author Response · Authors · 2025-11-27
> >
> > [continued]
> >
> > * **B1:** We completely agree with this point. We've addressed this concern above in our response to A2.
> > * **B2:** We appreciate this framing and agree that theoretical/methodological impact is important. To highlight this more clearly, we will revise the paper to emphasize three contributions that are *conceptual*, not merely empirical. We believe that these points are emphasized in our responses above. However, to summarize, our key contributions span methodological and pedological angles. (1)  A novel weighted Jaccard negative-pair formulation that creates a *compositional embedding geometry* (Sec. 3.2; Fig. 13 Appendix F). This is not used in prior composite restoration works. (2) A structured, interpretable latent geometry that enables controllability. Fig. 5 demonstrates emergent compositional interpolation, which is conceptually significant: unseen composites lie between their primitives. (3) A new real-world scientific utility benchmark includes remote sensing, microscopy, and urban forest monitoring. We *uniquely* show that these tasks require controllability and fidelity rather than aesthetics. We've revised the language in our contributions section to make these points more direct.
> > * **B3:** Thank you for raising this concern. We clarify that natural language prompting in PRISM is entirely optional. We have previously described the option for automated restoration via our CLIP-based multi-label distortion classifier, which removes the need for user input altogether. However, in a practical deployment, we imagine that a user may be able to interact with a standard GUI with clickable checkboxes that can serve as a precursor to (see the DiffPlugin demo [1] as an example). Natural language is provided as an *additional* modality for cases where experts must express fine-grained domain intent (e.g., “remove haze but preserve low-contrast structures”), which cannot be captured by a small set of buttons. Recent work has demonstrated that language can improve controllability and intent alignment in restoration and editing, starting with PromptIR (CVPR 2023) [2] and InstructIR (NeurIPS 2023) [3]. The diffusion methods we benchmark against follow similar structures for Stable Diffusion-based image- and text- guided restoration. The use of text here does not increase labor or unpredictability. It's meant to offer an optional, expressive channel for restoration.
> >
> > We thank the reviewer again for the thoughtful and rigorous critique. The new ablations, automation results, and clarifications should directly address all concerns raised. Please let us know if you have any further questions or concerns.
> >
> > References:
> >
> > [1] https://github.com/yuhaoliu7456/Diff-Plugin
> >
> > [2] Potlapalli, Vaishnav, et al. "Promptir: Prompting for all-in-one image restoration." Advances in Neural Information Processing Systems 36 (2023): 71275-71293.
> >
> > [3] Conde, Marcos V., Gregor Geigle, and Radu Timofte. "Instructir: High-quality image restoration following human instructions." European Conference on Computer Vision. Cham: Springer Nature Switzerland, 2024.

---

### Official Review · Reviewer_jZpm · 2025-11-02

**Soundness:** 2
**Presentation:** 2
**Contribution:** 2
**Rating:** 4
**Confidence:** 4

**Summary:**

This paper introduces PRISM (Precision Restoration with Interpretable Separation of Mixtures), a prompted conditional diffusion framework for expert-in-the-loop controllable restoration of images with compound degradations. PRISM integrates two key components: (1) compound-aware supervision trained on mixtures of distortions, and (2) a weighted contrastive disentanglement objective that aligns compound distortions with their constituent primitives. This approach enables high-fidelity joint restoration.

**Strengths:**

1. The paper is well-structured and the motivation is clear.
2. The framework leverages compound-aware supervision and a contrastive disentanglement objective across a diverse set of primitive tasks. This produces separable embeddings of distortions, enabling robust restoration, even for unseen real-world mixtures.
3. This work proposes a novel benchmark for scientific utility, spanning remote sensing, ecology, biomedical, and urban domains.

**Weaknesses:**

1. The method is devised on the basis of CLIP and Latent Diffusion Model. Moreover, Semantic Content Preservation Module is also relatively simple.
2. There is no physical information being incorporated into the training process. In other words, it is a general method that is used for scientific domains.
2. The dataset is constructed by integrating existing datasets, which is not very solid.

**Questions:**

1. Collecting a dataset from existing datasets is not very convincing. Have the authors collected any data?
2. It is a general method that is applied as a unified model for scientific and environmental images. More domain-specific priors, such as physical information, should be considered.
3. The text prompt is also short for benefiting these domains.

---

> ### Author Response · Authors · 2025-11-20
>
> We thank the reviewer for the constructive feedback and appreciate the opportunity to clarify our contributions. Below, we respond to each point.
>
> * As mentioned in "Re-Emphasizing Novelty & Contributions" in our Global Response, we wish to clarify that our primary contributions are not purely technical. They also lie in the **reframing of restoration as a controllable, compositional process and in the robust, benchmark-driven evaluation of that framing.** Specifically, PRISM posits a shift in perspective by prioritizing compound-aware restoration and prompt-guided controllability, two axes that are critical for real-world scientific applications but largely underexplored in prior work. To rigorously support this framing, we propose the MDB benchmark, which explicitly tests restoration fidelity under mixed and partial distortions with prompt supervision, and present a benchmark for evaluating the impact restoration on real-world downstream scientific tasks. Our semantic content preservation module is deliberately lightweight to preserve content fidelity without hallucination, which is critical in scientific domains. While simple, it substantially improves structure retention (see Appendix E for ablative results and Fig. 11 in Appendix F for additional details). PRISM's novelty lies in the compositional generalization of degradation-aware embeddings combined with prompt-driven, expert-controllable diffusion-based restoration, which has not been jointly-addressed in prior work.
> * We agree that physical priors are valuable, and we see our method as complementary to such approaches. However, our goal is to build a generalist system that can handle arbitrary mixtures across domains where explicit physical modeling is either impractical (e.g., microscopy) or unknown (e.g., drone-based reef imagery). We show in **Table 2 that PRISM generalizes across domain shifts and unseen distortions (UIEB, POLED, ThapaSet),** where physical modeling would require hand-engineered parameters per domain. In practice, many scientific users do not have access to detailed sensor models or calibration data. Thus, a learned compositional representation offers a practical path toward robust restoration under real-world complexity.
> * To the reviewer's point about our dataset construction, we respectfully clarify that PRISM introduces both (1) a new evaluation benchmark (MDB) constructed with compound-aware supervision, including synthetic and partially prompted examples to test selective restoration (Section 3.2) and (2) a new real-world dataset, the Rooftop Cityscapes benchmark, designed for evaluating restoration under urban degradations like haze and low-light (Section 3.2, Table 6). These **additions go beyond mere dataset aggregation.** The synthetic MDB is designed to probe compositional generalization under mixed and partial prompts, and we make our data augmentation pipeline open-source. Rooftop Cityscapes is manually curated from diverse lighting conditions with paired labels, enabling downstream evaluation (Table 3 & 4).
> * Our decision to use short prompts was intentional. While there are cases where a user may need to provide longer, descriptive instructions, in many scientific workflows such as camera trap analysis, domain experts provide brief correction requests. We design PRISM to operate effectively on short, natural prompts **(see Sections 3.1.1 and the prompting section)**, allowing for compositional control and interpretability. Moreover, we support partial prompts (targeting only a subset of distortions) and compositional generalization (Fig. 5), showing that even with short prompts, PRISM performs robustly. We **include additional ablations on the effects of variability and specificity in prompts in Appendix E.**
>
> Please refer to the updated PDF and supplemental for revised analysis and figures.

---

> ### Comment · Reviewer_jZpm · 2025-11-27
>
> Thanks for the detailed response. My questions have been addressed and have thus raised the rating to 6.

---

> > ### Author Response · Authors · 2025-11-27
> >
> > Thank you for taking the time to revisit our submission and for raising your rating. We appreciate your thoughtful engagement with the work and are glad the revisions addressed your concerns.

---

### Official Review · Reviewer_ztwF · 2025-11-02

**Soundness:** 3
**Presentation:** 3
**Contribution:** 3
**Rating:** 4
**Confidence:** 4

**Summary:**

This paper introduces PRISM, a controllable diffusion framework for compound image restoration, primarily targeting scientific applications. The method involves a two-stage process: first, fine-tuning a CLIP image encoder using a novel weighted contrastive loss to create a compositional embedding space for degradations , and second, training a latent diffusion model conditioned on these embeddings and user prompts to perform selective restoration.

**Strengths:**

The work compellingly argues for the necessity of controllable, selective restoration over automated 'full' restoration for scientific applications, demonstrating significant gains in downstream task utility.

**Weaknesses:**

The primary methodological concern is the limited novelty. The core idea of fine-tuning a CLIP encoder to be degradation-aware heavily relies on prior work (e.g., DA-CLIP). The main novelty appears to rest on the Jaccard distance weighting in the contrastive loss, but the paper lacks a direct ablation comparing this to an unweighted compound contrastive loss, making it difficult to isolate its true impact.

Second, the two-stage training pipeline  is computationally complex, and the choice of a diffusion backbone introduces significant inference latency. This efficiency trade-off is not sufficiently justified, especially as the performance gains over other recent diffusion methods  are notable but not transformative.

Finally, the framework's generalization to real-world, unseen degradations is questionable. The model is trained exclusively on synthetic composites , and it is unclear if the model is truly learning compositional physics or rather a powerful interpolation across its massive synthetic training domain when faced with the complex, non-linear physics of real-world degradations.

**Questions:**

What is the inference time of PRISM compared to the baselines (e.g., MPerceiver, AutoDIR, and the non-diffusion NAFNet), and how do the authors justify this computational cost for the observed performance gain?

Given the reliance on synthetic data , how can we be sure the model is learning robust compositional reasoning for real-world physics  rather than a complex interpolation?

---

> ### Author Response · Authors · 2025-11-20
>
> We thank the reviewer for the thoughtful and detailed critique. We address each concern below; please let us know if there are any further comments or concerns.
>
> * We acknowledge the reviewer's concerns about the novelty of the contrastive objective (see "Re-Emphasizing Novelty + Contributions" in Global Response). However, we wish to clarify that our primary contributions are not purely methodological. They lie in the reframing of restoration as a controllable, compositional process and in the robust, benchmark-driven evaluation of that framing. Specifically, PRISM posits a shift in perspective by prioritizing compound-aware restoration and prompt-guided controllability, two axes that are critical for real-world scientific applications but largely underexplored in prior work. To rigorously support this framing, we propose the MDB benchmark, which explicitly tests restoration fidelity under mixed and partial distortions with prompt supervision, and present a benchmark for evaluating the impact restoration on real-world downstream scientific tasks. Additionally, we show in Appendix E (Fig. 13) that the unweighted version collapses mixed-degradation representations, while our Jaccard-weighted variant preserves structure and improves partial prompting performance. **Fig. 4 and Appendix E Table 8 also ablates our model with and without contrastive reweighting,** and compares against other weighting schemes, demonstrating that this approach to restructuring the latent space, guided in principle by the challenges of compound degradations, improves overall performance and closes the gap between sequential and composite prompting. We have modified Section 4.1 with these details. While inspired by prior work on DA-CLIP, that line of work does not handle compound degradations or prompt-driven control, nor does it model compositional alignment of compound degradations. Thus, **PRISM's novelty lies in the compositional generalization of degradation-aware embeddings combined with prompt-driven, expert-controllable diffusion-based restoration.**
> * As discussed in the **"Efficiency and Computational Costs of PRISM" comment in the Global Response,** PRISM incurs similar costs to similarly-performing diffusion models in this space. We intentionally center our approach on diffusion: as shown in Table 1 and Table 2, diffusion-based baselines (AutoDIR and MPerceiver) significantly outperform other methods on compound and real-world degradations. PRISM builds on this strength and adds both automated distortion detection and compositional control, matching or exceeding all baselines on both fidelity and perceptual metrics (Tables 1). We note that PRISM also achieves significant gains on unseen combinations of distortions (Fig. 3 and Table 2), uniquely supports selective restoration (Tables 3 and 4), prompt-driven compositional control (Fig. 5), and interpretability via disentangled embeddings (Fig. 4) which none of the baselines offer all at once. We believe the modest increase in compute is well-justified for practical applications where handling real-world mixtures and expert input are a priority, not computational cost. PRISM demonstrates that expert-guided restoration can be realized without sacrificing practical deployability. For better transparency, we have inserted a high-level summary of these results to the main text near the conclusion.
> * The reviewer's comment about PRISM's generalization to real world degradations and physical modeling is well-received. We clarify that *we do not claim that PRISM models physical degradations explicitly.* Rather, it learns compositional latent representations of distortions from data. We **revise the experimental design and language in Sec. 4.2** to qualify these claims. Our **results in Table 2 on datasets featuring complex, nonlinear, real-world degradations** never seen in training (UIEB, POLED, ThapaSet) demonstrate state-of-the-art zero-shot generalization after automatically detecting the most relevant primitives, outperforming both diffusion and non-diffusion baselines. For instance, on ThapaSet, which includes nonlinear fluid distortions, PRISM generalizes robustly, despite no explicit physics supervision. This suggests that PRISM is not merely interpolating over synthetic textures, but has learned a structure-preserving embedding space where unseen combinations can be decomposed and corrected.
>
> Please refer to the updated PDF and supplemental for revised analysis and figures.

---

### Official Review · Reviewer_S4VD · 2025-11-02

**Soundness:** 3
**Presentation:** 2
**Contribution:** 3
**Rating:** 4
**Confidence:** 3

**Summary:**

The paper proposes a framework for controllable restoration of images that underwent multiple primitive distortions. The restoration of the degradations happens at once, rather than consecutive restorations that may introduce artifacts, yet enables defining which degradation to restore in order to preserve necessary signals.

As part of the training process, a restoration benchmark with tuples of a clean and a degraded image, along with a natural language prompt that describes the degradation is used, which is made public. Given a prompt, the framework includes using a frozen CLIP text encoder for multi-label classification from a set of primitive degradation, which are then formatted to a predefined form of prompt. This prompt is then used to restore the image by applying a finetuned version of SD 1.5, where the CLIP image encoder was finetuned to cluster embeddings by degradation, followed by an additional model trained to preserve semantic content.

The overall framework is evaluated on a benchmark with images that underwent distortions as the ones in the training process (up to 3 primitive distortions) and on unseen distortions. Furthermore, its usage for four downstream tasks is evaluated, showing scientific utility.

**Strengths:**

* Image restoration is a critical task, particularly for scientific applications. This paper demonstrates the method's effectiveness through general purpose image restoration, evaluated using fidelity and perceptual metrics, and its application for downstream scientific tasks.
* The motivation is written clearly, and the figures (although 1 and 2 are not referenced) support the understanding of the general approach.
* Although the number of consecutive distortions in the training set is limited to 3, section 4.2 shows that the method archives good results also for unseen degradations that are not necessarily a combination of the degradation in the train set, or are a combination of more than 3 primitive distortions.

**Weaknesses:**

* While the fine tuning of CLIP image encoder is explained thoroughly, the following steps of how SD 1.5 is used as the backbone and the suggested SCPM module are explained only briefly. This impairs the understanding of the entire framework, and while the code is submitted, the text itself is not sufficient to reproduce the code.
* The concept of automatic restoration needs clarification. While the paragraph on prompting (line 207) describes the automatic transformation of natural prompts to fit the required format, the method for generating these automatic prompts remains unclear. Possibly related, it is unclear which prompts were used in the experiments in Sec. 4.1.
* The motivation of controlled restoration for expert in the loop scenarios could be further evaluated by comparing images that underwent multiple distortions but PRISM is prompted to restore only a partial set (as in Sec. 4.3.1 on synthetically) where the control of the restorations suggests different restorations for specific images / use cases rather than a predefined subset for all images in the same domain.

**Minor:**
* Indices are not explained and somewhat confusing. It seems $d_{i_j}$ in Eq. 1 denotes a specific distortion and $d^i$ in row 180 denotes a set of distortions, yet the notations are explained only after being used ($d_j$ is explained in line 200).
* Using the Jacard distance between degradation sets neglects how some distortions are more similar than others.
* Should mention how the prompts in the dataset are auto generated (line 157).

**Questions:**

* In addition to PSNR reported in Fig 3, what was the effect on other discussed metrics?
* What value is used for the number of variants $m$ ? And what is the minibatch size? If these values are not similar there might be added values in weighing the two terms in the denominator of the per-variant contrastive loss to control the effect of repelling from other degradations and that of repelling from other images.
* Is there a difference between the dataset described in Sec.3.1 and the benchmark described in 3.2 (MDB)? If so, what is included in MDB?
* In Sec. 4.2, did PRISM identify the same set of primitive distortions for different images from each domain where images probably went through similar degradation?
* Which images were used to create the visualization of the scatter plot in Fig. 5?

---

> ### Author Response · Authors · 2025-11-20
>
> We thank the reviewer for the careful reading and helpful suggestions. Below we respond point-by-point to the technical and conceptual concerns raised; please let us know if there are any further comments.
>
> * We agree the main text needed clearer scaffolding and now explicitly reference specific implementation details in the Methods. We’ve **clarified implementation details in Section 3.2.2,** including explicit use of Stable Diffusion v1.5 and added a detailed description of the Semantic Content Preservation Module (SCPM) under "Refining Outputs." The SCPM ablation is in Appendix E, with implementation details in Appendix F (Fig. 12).
> * To clarify, all prompts (including compound, partial, and negative examples) were automatically generated using GPT-4, sampling diverse phrasings per distortion (**see lines 156 - 165 in Section 3.1, and Appendix B for precise methodology and queries used**). Section 3.3 now outlines the two primary prompting mechanisms PRISM enables: (1) manual inputs specified by experts and (2) automated prompt construction. As described in Section 3.3 (lines 256-263), PRISM can predict distortions from the fine-tuned image embedding and use this prediction to generate structured prompts. Lastly, to clarify, **in Section 4.1, manual prompting from our set of degradation-specific prompts** was used (this was initially stated in lines 321-323, and we have now specified this throughout Section 4.1).
> * While we appreciate the reviewer’s suggestion to further motivate controllability through synthetic or simulated examples, we respectfully argue that our motivation for controllability is rooted in downstream use (Sec. 4.3.1). We agree that additional synthetic simulations could serve as illustrative case studies, but we believe the current formulation (Section 4.3.1) provides a more rigorous, task-relevant benchmark for controllability in practice. Separately, we design the MDB benchmark as a held-out test set that includes both full, partial, and negative degradation cases, as in the training dataset. This enables direct assessment of whether models can follow selective prompts (i.e., removing haze while preserving other distortions). In Appendix E Table 9, we report that PRISM achieves 87.7% prompt faithfulness (proportion of outputs that correctly follow the instruction) on this test set. **Appendix E also contains expanded sensitivity analysis on the flexibility of prompts** (specificity, variation in language, etc.)  that shows how PRISM's performance is invariant to reasonable variations in the language of the instructions. This result supports our claim that PRISM supports fine-grained controllability and can accurately follow compound and selective instructions.
> * We’ve fixed notational inconsistencies in the contrastive loss equations in Section 3.1.1. Thank you for pointing this out!
> * We agree Jaccard distance is a simplification. Our goal was to encode compositional alignment, not perceptual similarity. Jaccard similarity serves as a proxy for degradation set overlap, which is sufficient for modeling compound structure based on our results. Future work may incorporate richer priors.
>
> Please refer to the updated PDF and supplemental for revised analysis and figures.

---

> ### Author Response · Authors · 2025-11-20
>
> Here, we address the specific questions raised by the reviewer.
>
> * **Appendix F Figs. 14–15 now include SSIM and FID** (alongside PSNR and LPIPS). We update Section 4.1 with pointers to these results, which confirm PRISM's gains across fidelity and perceptual metrics, especially under compounding distortions.
> * We thank the reviewer for this insightful comment regarding our loss. We've **clarified these parameters to Section 3.2.1.** While our use of uniform weighting yields strong results, future work could explore adaptive schemes.
> * Thank you for noting the ambiguity between the training dataset and the MDB benchmark. To clarify, Section 3.1 describes our training set, which includes distorted variants with up to three compound degradations and full, partial, and negative prompt supervision. Section 3.4 describes our MDB evaluation benchmark, containing distorted examples held out during training, paired with clean references and prompts. While the MDB set uses the same basis of primitive distortion types, it includes also unseen combinations of up to four distortions and is designed to test compositional restoration and prompt faithfulness. We have also **edited the analysis in Section 4.1 to clarify the specific evaluations run on MDB.** In precise terms, the MDB evaluation set contains images with up to three distortions, and we further assess generalization to more complex, unseen combinations of four distortion types applied to the same set of images in the MDB.
> * In our zero-shot tests, the same set of manually specified primitive prompts were used per domain for standardized experimentation. PRISM's ability to restore these domains arises from its latent compositional structure (see Fig. 5), not memorization. We have **clarified this in Section 4.2.**
> * We’ve **updated captions for Fig. 5 and Appendix Fig. 12** to clarify that embeddings are UMAP projections of CLIP-space features over MDB samples (clean + distorted). Thank you for pointing this out!
>
> As mentioned above, please refer to the updated PDF and supplemental for revised analysis and figures.

---

> > ### Comment · Reviewer_S4VD · 2025-11-27
> >
> > I thank the authors for their detailed rebuttal and revisions which addressed my main concerns.
> >
> > Regarding the motivation of controlled restoration for expert in the loop scenarios, my initial understanding was that an expert could apply different restorations on a given image (or set of images), that intentionally don’t try to restore all aspects of degradation. As I understood the motivation, the expert does not necessarily know beforehand which is the best set for his downstream task. This motivates having a model which can be prompted more than once, using different combinations, to get the best results for a specific case.
> > For this reason, applying the same predefined set for all images in the same domain as done in Sec. 4.3.1 is somewhat weaker than e.g. showing how images from the same domain should be restored with different sets when aiming for different downstream tasks. As if there is a “ground truth” set for each domain, it might be better to have a model per domain and the speciality of PRISM by being able to apply many combinations is not exploited.

---

> ### Author Response · Authors · 2025-11-27
>
> Thank you for the thoughtful follow-up! Your observation is exactly right, and we appreciate the opportunity to clarify this more explicitly. Our emphasis on **scientifically motivated controllability** stems from the fact that in many real analysis pipelines, the optimal restoration is not known a priori and is often **task-dependent even within the same domain**. This is precisely the scenario where PRISM’s compositional prompting is uniquely useful. Your concern is well-taken: in Sec. 4.3.1 we used a standardized distortion subset per domain, and this choice may have unintentionally obscured PRISM’s ability to explore **multiple restoration combinations for the same image**, depending on the downstream task.
>
> To directly address this, we have made the following clarifications and additions:
>
> * **Why we standardized prompts in Secs. 4.2 and 4.3.** These experiments aimed to compare models fairly under fixed, predefined restoration categories, some of which are required by baselines. This ensured controlled comparisons, but we understand that this may suggest that a domain has a “ground truth” distortion set. We agree that this setup does not fully showcase PRISM’s flexibility. To address this, we have removed the results in 4.3, and modified the experiment in 4.2 to use PRISM's automated distortion classifier to establish the set of distortions *per image* to remove. This improved the margins over the baselines!
> * **Explicit clarification of our motivation.** We now state clearly in Sec. 4.3.1 (now 4.3) that **different downstream tasks on the same domain often require different restoration subsets**, and that this variability is a central motivation for PRISM. We emphasize that the per-domain prompt choices were illustrative, not definitive.
> * **New experiment demonstrating task-dependent restoration.** We thought about this point very carefully when designing our experimental design for Sec. 4.3.1 (now 4.3), and had also considered looking at different downstream tasks for the microscopy data. To directly support your point, we added an experiment to this analysis showing that on microscopy images, segmentation benefits most from **super-resolution** while fluorescence quantification benefits most from **denoising** and combined restoration can hurt both tasks. Since these are two tasks performed on the same raw images, this result concretely demonstrates that **no single restoration set is optimal per domain**, and PRISM’s controllability is essential to explore and select the appropriate subset. We hope that this directly addresses the reviewer's concern.
> * **Real-world evidence of per-image adaptation.** To provide more concrete evidence of PRISM's flexible restoration, in Appendix Figs. 20–21, we show real examples where PRISM automatically predicts and restores different degradations for different images within the same dataset. For instance, certain camera trap images benefit from contrast correction while others require motion-blur removal. These examples reinforce that, as you've already noted, PRISM already performs **per-image, non-uniform restoration**, even before introducing task-level variation.
> * **Revised framing in the abstract, introduction, and results.** We updated these sections to emphasize that PRISM supports both **expert-guided selection of restoration subsets** and **automated inference of distortions**, without requiring per-domain retraining. The compositional latent space enables precisely the multi-combination exploration you describe.
>
> With the new task-dependent experiment and revised framing, we hope it is now more evident how PRISM’s controllability is not only scientifically-motivated but often necessary. Please let us know if you have any further questions or concerns!

---

### Official Review · Reviewer_atD5 · 2025-11-04

**Soundness:** 3
**Presentation:** 3
**Contribution:** 3
**Rating:** 6
**Confidence:** 3

**Summary:**

The paper presents PRISM, a prompted, controllable diffusion framework for restoring images suffering from compound degradations. The training setup uses mixtures of up to three distortions and introduces a weighted contrastive disentanglement stage to make embeddings separable and compositional. Inference accepts free-form prompts that are mapped to a fixed set of restoration labels; a latent-diffusion backbone performs restoration and a Semantic Content Preservation Module (SCPM) refines fine detail. Experiments cover a new Mixed Degradations Benchmark (MDB), zero-shot evaluation on real datasets (UIEB, under-display camera, and fluid lensing), and downstream tasks across remote sensing, ecology, microscopy, and urban scenes.

**Strengths:**

1) Clear objective and method design. The paper argues for simultaneous rather than sequential restoration, emphasizes expert control, and focuses on scientific fidelity rather than aesthetics. The architecture coherently combines contrastive disentanglement, prompt-conditioned latent diffusion, and SCPM for detail recovery.
2) Good reported performance and breadth. On MDB, PRISM outperforms representative all-in-one and diffusion/composite baselines (e.g., AirNet, Restormer, NAFNet, PromptIR, OneRestore, DiffPlugin, MPerceiver, AutoDIR) on PSNR/SSIM and perceptual metrics.
3) Generalization beyond the synthetic training setup. The paper reports zero-shot gains on real distortions (underwater, under-display camera, and fluid distortions) and shows that performance scales well as the number of simultaneous degradations increases.
4) Practical value for scientific analysis. Selective, prompt-guided restoration improves downstream tasks in several domains, supporting the utility of controllability.

**Weaknesses:**

1) Control granularity and evaluation scope:
  The evaluation largely uses manual prompting with a pre-defined set of distortion types, not open-ended language or fine-grained controls. The paper itself notes that extending controllability beyond “which distortions to remove” to specifying intensity and spatial extent is left for future work. This leaves unanswered how robust the system is to realistic prompt variations or local/severity-aware edits.

2) Synthetic-to-real gap and capped composition complexity:
   Training relies on synthetic mixtures and explicitly caps each sample at up to three distortions for efficiency and interpretability. The authors acknowledge that these synthetic augmentations cannot fully capture real distortions. While results on real datasets are encouraging, this cap and reliance on synthetic compositing may underrepresent harder real-world compound effects.

3) Efficiency and deployability are not quantified in the main text:
   The paper does not provide main-text wall-clock, throughput, or memory comparisons versus strong diffusion baseline. Without standardized timing/FLOP/peak-memory profiles at a fixed resolution, it is difficult to assess practical deployability or the overhead of the added control and SCPM modules.

**Questions:**

.

---

> ### Author Response · Authors · 2025-11-20
>
> We thank the reviewer for the thoughtful feedback. Below, we address each of the three primary concerns raised; please let us know if there are any further comments or concerns.
>
> * We acknowledge that enabling open-ended and fine-grained control in the restoration prompts (such as specifying severity or spatial extent) is an important next step. However, we emphasize that PRISM already supports *non-trivial controllability* that extends beyond most prior work through training and evaluating on examples of selective restoration (only removing a subset of distortions) and prompts that capture the open-ended variation in natural language commands (we generate diverse phrasings with GPT-4, see the updated Section 3.1 and Appendix B). The reviewer commented that the our presented results primarily target manual prompting with a pre-defined set of distortion types. We want to note that although this prompting was with a pre-defined set of distortions, this standardization was intentional to avoid any confounding impacts of automatically detecting distortions. We have revised our experiment on the zero-shot performance of PRISM to use automated restoration. This highlights that our method not only enables automated detection of primitive distortion types, but can restore with higher fidelity compared to baselines. To assess PRISM's robustness to realistic prompt variations, we ran sensitivity analysis over different prompting styles (over-specific, vague, incorrect, incomplete, varied phrasing/vocabulary, etc.) in Appendix E Tables 11 and 12. Results show high consistency in PSNR, SSIM, and LPIPS across all variants, indicating strong robustness to linguistic variability. **Fig. 11 (see Appendix E, under "Sensitivity to Local and Intensity-Aware Prompts")** shows qualitative examples showing varied levels of restoration under varied severity levels. While we do not yet support continuous severity "sliders" or location masks, the current setup already enables meaningful expert-in-the-loop controllability and reasonably performs well under non-standard prompt formats, as qualitatively demonstrated **real-world use cases in Figs. 5 and 6** in the main text (e.g., "fix coloring," "unblur", "super-resolve").
> * We appreciate the reviewer's concern about the synthetic-to-real gap and agree that reliance on synthetic data augmentations introduces inherent limitations. Our training distortions were designed to span 14 distinct geometric, photometric, occlusion-based, and noise-based degradation types, covering a wide range of real-world effects (Appendix B). While each training image contains up to three composited distortions, PRISM's strength lies learning compositional structure over a latent space of degradations. This enables zero-shot generalization to domains like underwater imaging, fluid lensing, and under-display cameras (Table 2), where distortions differ physically but align geometrically with trained primitives (see Fig. 5). We agree that certain distortions may fall outside this learned space. We analyze one such **generalization failure in Appendix F (Fig. 22, bottom left)**, where a digital moiré effect produces structured, high-frequency artifacts not encountered in our training set. While PRISM does not fully remove this distortion, it still manages significant qualitative improvements, suggesting a degree of robustness even in underrepresented regimes. These results provide a proof-of-concept for compositional alignment as a scalable mechanism where new degradations can be embedded into the learned latent space and interpolated against known primitives for open-ended generalization. Extending this to support harder, higher-order mixtures remains a promising direction for future work.
> * We agree that computational costs are an important consideration. **Table 7 in Appendix Section E (under "Cost and Latency Analysis")** summarizes inference benchmarks (FLOPs, memory usage, runtime) on an NVIDIA A100 GPU at $256 \times 256$ resolution. Despite incorporating the SCPM and the distortion classifier, PRISM achieves similar FLOPs, memory usage, and latency as other high-fidelity Stable Diffusion-based image editing methods such as DiffPlugin, MPerceiver, and AutoDIR. As mentioned in the global response, we include a high-level summary of this comparison in our discussion for better transparency.
>
> Please refer to the updated PDF and supplemental for revised analysis and figures.

---

### Author Response · Authors · 2025-11-20
**Global Response to All Reviewers**

We thank the reviewers for their insightful, thoughtful feedback! We appreciate that they found that:

* The paper's **framing was novel** and the **motivation was clear** (reviewers atD5, S4VD, jZpm, and dhPX).
* The approach was well-structured and **results were strong**, particularly in the generalization to complex or unseen test cases (reviewers atD5, jZpm, and S4VD).
* The **evaluation framework was diverse and strong**, with the construction of a novel benchmark dataset (reviewers atD5, S4VD, jZpm, and dhPx).

Below, we highlight key clarifications/revisions, and include reviewer-specfic repsponses to address remaining feedback. We refer the reviewers to our updated PDF and supplementary materials for revised analysis and figures. Any major **updates are highlighted in blue** in our revised PDF.

## Re-Emphasizing Novelty & Contributions
While PRISM builds on CLIP and latent diffusion, our core contribution lies not only in technical novelty, but in reframing restoration as a controllable, compositional process for scientific use cases. This framing emphasizes the importance of prompt-guided controllability for selective restoration, compound-aware supervision for real-world mixed distortions, scientific fidelity over aesthetics. PRISM's contributions lie in the compositional generalization of degradation-aware embeddings combined with prompt-driven, expert-controllable diffusion-based restoration, which have not been jointly addressed in prior work. We also introduce a novel benchmark over real-world use cases of restoration in diverse scientific domains, and unique nuanced analysis on the importance of controllability guided by downstream performance. We **clarified this framing in our discussion, and revised wording to better distinguish our contributions from prior work.** Some reviewers commented that PRISM's performance gains on traditional benchmarks or standard metrics appear modest. We agree this is true in low-complexity settings (e.g., single distortions), where existing baselines are strong. However, we emphasize that **PRISM particularly excels under compound and unseen degradations** (Table 2, Fig. 3), which are especially relevant to scientific deployment. We have clarified this point in our analysis and **added references to SSIM and FID scores in Appendix F** to show that improvements generalize across metrics.

## Controllability for Precision
Multiple reviewers expressed concern around the paper’s use of the term “precision,” particularly regarding whether it refers to prompt controllability or output fidelity. To clarify, we do not conflate controllability with precision. Rather, we argue that controllability (the ability to specify selective restoration) is a mechanism that *enables* precision in scientific settings, where over-restoration risks suppressing subtle but critical features (e.g., in microscopy or remote sensing). **We have revised the framing and clarified this in Section 2.1 and Section 4.3.1.** The reviewers requested stronger or more illustrative evaluations of PRISM’s controllability and prompt adherence. We appreciate this feedback and have **expanded our evaluation of sensitivity to prompt variations,** which includes incorrect, incomplete, vague, or over-specific prompts. These changes aim to strengthen the rigor and clarity of our controllability evaluation. While we emphasize the importance of controllability and selective restoration, we also modified the text to clarify **PRISM's two-fold restoration: (1) prompted and (2) fully-automatic.**

## Efficiency and Computational Costs of PRISM
While diffusion-based restoration models inherently incur greater computational cost than encoder–decoder baselines, we find that PRISM maintains competitive runtime efficiency relative to comparable diffusion frameworks. **Table 8 in Appendix Section E (under "Cost and Latency Analysis") summarizes inference benchmarks (FLOPs, memory usage, runtime)** on an NVIDIA A100 GPU at $256 \times 256$ resolution. Despite incorporating the lightweight distortion classifier for automated restoration and the SCPM module, PRISM achieves similar FLOPs, memory usage, and latency as other Stable Diffusion-based image editing methods such as DiffPlugin, MPerceiver, and AutoDIR. That said, we argue that in many scientific workflows, inference time and computational cost are not the primary bottlenecks, the inability to handle real-world mixtures, black-box pipelines, and low downstream fidelity are. **We have updated our discussion at the end of the paper to reflect our analysis on the tradeoffs between latency and fidelity.**

---

> ### Author Response · Authors · 2025-11-20
> **Minor Revisions**
>
> In addition to the main revisions/responses outlined above, we would also like to highlight some of the smaller revisions that were not mentioned in our previous response:
> * We added further details on the evaluation datasets used for each set of results presented throughout Section 4.
> * Some reviewers raised questions about the distortion types used for training, particularly the use of 3-way compositions and whether this threshold was empirically or arbitrarily chosen. We have clarified in the main text that this decision was based on (i) realism (most scientific degradations exhibit 2–3 co-occurring effects), and (ii) stability (empirical tests show performance degrades when training with more than 3 distortions.) These results are now discussed in Appendix E (“Role of Distortion Count in Data Augmentation Pipeline”).
> * Appendix E now includes ablations on distortion taxonomy and analysis on prompt supervision levels.
> * We revised the title to make it more easy-to-read and engaging.

---

### Author Response · Authors · 2025-12-03
**Summary of Rebuttals**

# Summary of Rebuttals

We thank the reviewers for their careful evaluation of our submission and for the constructive dialogue. We also appreciate the AC’s engagement in light of the challenging review situation. In this comment, we summarize (1) the main points raised across reviews, (2) how we addressed them through revisions, new analyses, and clarification, and (3) key takeaways and why we believe the work is now in a strong position for acceptance.

## 1. Summary of Reviewer Feedback

Here we highlight the overall strengths and primary concernes raised by the reviewers.

### Strengths

* **Clear motivation and novel framing** for expert-in-the-loop, controllable restoration (atD5, S4VD, jZpm, dhPx).
* **Strong results on compound and unseen real-world degradations**, with broad domain coverage (atD5, S4VD, ztwF).
* **Value of selective restoration** for scientific downstream tasks (atD5, ztwF).
* **Quality and novelty of the benchmark design** and the substantial engineering effort (atD5, S4VD, jZpm, dhPx).

### Primary Concerns

* **Novelty and clarity of methodology**, especially the contrastive loss, SCPM, and SD-based pipeline (S4VD, ztwF, jZpm).
* **Sensitivity to prompts**: extent of control, prompt variability, and whether controllability truly improves precision (atD5, S4VD, dhPx).
* **Generalization** from synthetic data and whether the model learns meaningful compositional structure (atD5, ztwF, dhPx).
* **Efficiency, scalability, and complexity** of the two-stage architecture (atD5, ztwF).
* **Choice of distortion count (≤3)** and justification for this limit (atD5, S4VD, dhPx).
* **Weak results on downstream task baseline**, which makes sense since some of the compared models are intrinsically more robust to noise.
* **Role of expert prompting vs. automation**, and whether prompts impose unnecessary human burden (dhPx).
* **Dataset construction and prompt generation clarity** (S4VD, jZpm).

---

> ### Author Response · Authors · 2025-12-03
>
> ## 2. Summary of Our Responses and Revisions
>
> Across rounds of discussion, the authors provided detailed clarifications and extensive new experiments. **Notably, reviewer jZpm raised their score** after the rebuttal, writing that their concerns had been fully addressed. Key resolutions include:
>
> ### Automated vs. Manual Prompting (from reviewer dhPx's follow-up)
>
> * Clarified that PRISM already includes a **fully automated restoration mode** using the CLIP-based multi-label classifier (added Sec. 3.3), with **no human input required**. We didn't emphasize this point in the initial submission in favor of highlighting the importance of selective restoration, but we have woven this contribution throughout the text (Abstract, Introduction, etc.)
> * We **modified how automated instructions are displayed** in our paper's figures to make this distinction more clear.
>
> ### Prompt Robustness
>
> * Added **new analyses** showing that PRISM is highly faithful to the input instructions and remains stable under *incorrect, vague, or varied* natural-language prompts (Appendix E).
> * Added experiments comparing **automated, fixed, and free-form prompting**, showing that PRISM is *not* dependent on human decisions (Appendix E).
> * Introduced **new ablations on prompt granularity and accuracy**, directly addressing reviewer dhPx’s concerns.
> * Added **intensity-aware and locality-aware prompt qualitative studies**, showing the model’s limited sensitivity on these specific prompts.
>
> ### Synthetic-to-Real Generalization
>
> * Strengthened analysis explaining that PRISM does not model physics explicitly, but learns a **compositional latent geometry** enabling zero-shot generalization. We now demonstrate in Figure 5 how this supports both automated and iterative restoration.
> * **Modified analysis on zero-shot datasets** UIEB, POLED, and ThapaSet to first detect which distortions are present and then restore over that standardized set across baselines. This demonstrates not only stronger generalization via primitive distortion detection but also higher-fidelity performance exceeding state-of-the-art diffusion and non-diffusion baselines (Table 2).
>
> ### Distortion Count Justification
>
> * Added a more thorough empirical study showing that training on >3 distortions degrades embedding separability, reduces restoration fidelity, increases prompt ambiguity, and does not reflect real-world mixture statistics. See Appendix E: “Role of Distortion Count in Data Augmentation Pipeline”. This directly addresses concerns from atD5, S4VD, and dhPx.
>
> ### Methodological Clarity & Novelty
>
> * Expanded Sec. 3.2.2 to provide **clearer implementation details** for the diffusion backbone and the SCPM module.
> * Clarified conceptual contributions beyond architecture to emphasize compositional embedding geometry, controllable diffusion for scientific fidelity, and our new scientific utility benchmark.
> * We've **clarified dataset descriptions**, benchmark construction, and real-world domain results.
> * Included more details Appendix to **improve reproduciblity** (included GPT-4 queries for generating prompts, random seeds, etc.).
>
> ### Efficiency & Computational Costs
>
> * Added a new latency/memory/FLOP comparison (Appendix E), showing PRISM has **comparable runtime** to diffusion-based baselines (DiffPlugin, AutoDIR, MPerceiver) despite additional modules.
> * Added summary discussion in the main text highlighting this comparison.
>
> ### Controllability vs. Precision (from reviewer S4VD's follow-up)
> * **Clarified framing** of controllability vs. precision (Sec 4.2, Sec. 4.3).
> * Added a new experiment in Appendix E showing that **different downstream tasks on the same data require different restoration subsets**, validating PRISM’s motivation and demonstrating real scientific necessity for controllability. (review S4VD follow-up)
> * **Removed results from Section 4.3** that were not statistically meaningful, replacing them with this stronger and more principled experiment.
>
> ### Miscellaneous Edits:
> * Added **ablation over reweighting strategies** in Appendix E Table 8.
> * **Expanded results** of SSIM/FID added to the extended versions of Figures 3 and 4 (Appendix F).
> * **Refined notations, figures, and captions**, and reorganized explanations for clarity (notably prompted by S4VD).
> * **Minor presentation improvements**, including a clearer title and clearer language.
> * We report **classification performance** over our MDB set with the distortion detector over the fine-tuned CLIP embeddings.
>
> All major updates are highlighted in blue in the revised PDF. We note that previous responses may refer to figres/sections or line numbers that have been shifted or removed entirely from the current submission.

---

> > ### Author Response · Authors · 2025-12-03
> >
> > ## 3. Key Takeaways
> >
> > This submission now presents a **well-motivated advancement** in image restoration for scientific applications. PRISM offers:
> >
> > * A **novel compositional framework** enabling joint restoration of compound disturbances.
> > * **Selective, controllable restoration**, which we demonstrate as essential in scientific domains where over-restoration can degrade data utility.
> > * **State-of-the-art performance** on complex mixtures and **strong zero-shot generalization** to challenging real-world data.
> > * A **new benchmark suite** that brings scientific utility—a crucial but underexplored dimension—into the evaluation of restoration models.
> > * Thorough, transparent ablations and analyses that directly address reviewer concerns and clarify both the method's novelty and its practical impact.
> >
> > The value of this work lies in the substantial engineering effort invested in dataset reconstruction and the novel perspective it brings to scientific image restoration under complex degradations. We include a demo website to more easily interact with our figures and model outputs: https://prismrestore.github.io/. Given the strengthened analyses, the expanded experiments, the clear scientific motivations, the positive response from multiple reviewers (including an explicitly raised score), and the significant revisions made in response to feedback, we believe the paper is now **substantially improved and well-positioned for acceptance**.

---

### Meta-Review · Area_Chair_Q6TD · 2026-01-06

**Summary:**

This paper presents PRISM, a framework designed for controllable and compositional image restoration in scientific domains. By combining a weighted contrastive loss with a CLIP-guided diffusion backbone, the method enables the selective disentanglement of compound degradations. The authors also contribute a new benchmark (MDB) to evaluate restoration fidelity. The review process was active and productive. While initial reception was mixed (ratings ranging from 2 to 6), the rebuttal significantly strengthened the paper, particularly by clarifying the scientific necessity of "human-in-the-loop" control versus black-box automation.

**Reviewer Concerns:**

Addressed Concerns: The most critical debate centered on the motivation for "controllability." Reviewer dhPx argued that introducing expert prompting adds unnecessary labor compared to full automation, while Reviewer S4VD initially questioned the practical utility of selective restoration. The authors provided a compelling response: they not only clarified that PRISM already supports a fully automated mode but, more importantly, demonstrated that optimal restoration is task-dependent. For instance, they showed that downstream tasks like segmentation versus fluorescence quantification on the same microscopy image require different restoration subsets. This effectively settled the debate, proving that controllability is a requirement for scientific rigor rather than a methodological limitation. Additionally, concerns regarding dataset construction (jZpm) and generalization to real-world physics (ztwF, atD5) were satisfactorily resolved through clarifications and strong zero-shot experiments on datasets like UIEB and ThapaSet.

Minor Concerns:
Some limitations remain. Reviewer ztwF noted that the technical core (CLIP + Stable Diffusion) relies heavily on existing components. I agree the architectural novelty is somewhat incremental; however, the primary value here lies in the rigorous system design and framing tailored for scientific fidelity, which is a significant contribution in itself. Additionally, the reliance on diffusion models introduces inference latency compared to regression baselines, a known trade-off that the authors reasonably justify for high-stakes, offline scientific analysis.

**Reviewer Scores:**

Reviewers atD5 and jZpm support acceptance (Score: 6), citing strong motivation and the resolved data concerns.
Reviewer S4VD (Score: 4 -> 6) reacted very positively to the new task-dependent experiments during the discussion.
Reviewer ztwF (Score:: 4) remains borderline due to novelty concerns but acknowledges the robust evaluation.
Reviewer dhPx (Score: 2) kept a lower score, suggesting that the method should ideally be fully automated. However, in light of the new experiments showing that different tasks require different restoration strategies for the same image, I find the authors' justification for a 'human-in-the-loop' design to be empirically sound.

---

### Decision · Program_Chairs · 2026-01-26

Accept (Poster)